# A neural network model of hippocampal contributions to category learning

Jelena Sučević[1], Anna C Schapiro[2]*

[1]Department of Experimental Psychology, University of Oxford, Oxford, United Kingdom; [2]Department of Psychology, University of Pennsylvania, Philadelphia, United States

**Abstract** In addition to its critical role in encoding individual episodes, the hippocampus is capable of extracting regularities across experiences. This ability is central to category learning, and a growing literature indicates that the hippocampus indeed makes important contributions to this form of learning. Using a neural network model that mirrors the anatomy of the hippocampus, we investigated the mechanisms by which the hippocampus may support novel category learning. We simulated three category learning paradigms and evaluated the network's ability to categorize and recognize specific exemplars in each. We found that the trisynaptic pathway within the hippocampus—connecting entorhinal cortex to dentate gyrus, CA3, and CA1—was critical for remembering exemplar-specific information, reflecting the rapid binding and pattern separation capabilities of this circuit. The monosynaptic pathway from entorhinal cortex to CA1, in contrast, specialized in detecting the regularities that define category structure across exemplars, supported by the use of distributed representations and a relatively slower learning rate. Together, the simulations provide an account of how the hippocampus and its constituent pathways support novel category learning.

## Editor's evaluation

This article will be of interest to a broad audience of cognitive neuroscientists interested in learning and memory, especially those who study the computations of the hippocampus in human and animal models. This work offers compelling evidence in support of a role for the computations theorized to occur within the hippocampus in category learning more generally. The well-conducted and rigorous computational simulations support the key conclusions and offer a novel theoretical entry into characterizing human learning.

## Introduction

Learning how the entities in our environment cluster into groups with overlapping properties, names, and consequences allows us to communicate and act adaptively. This category learning often unfolds over long periods of time, for example, when learning about the many species of dogs across development, but can also occur within a few minutes or hours, as when learning about different kinds of penguins on a first visit to the zoo. Much is known about how neocortical areas represent categories of information learned over long timescales (*Martin, 2007*; *Miller et al., 2003*), but less is understood about the mechanisms by which the brain learns quickly in initial encounters. Given the ability of the hippocampus to learn rapidly (*McClelland et al., 1995*) combined with its ability to learn regularities across experiences (*Schapiro et al., 2012*), this brain area seems well suited to make a contribution to rapid category learning (*Mack et al., 2018*; *Zeithamova and Bowman, 2020*). Indeed, neuroimaging studies provide strong evidence that the hippocampus is engaged in novel category learning (*Bowman and Zeithamova, 2018*; *Mack et al., 2016*; *Zeithamova et al., 2008*). Studies

*For correspondence:
aschapir@sas.upenn.edu

with hippocampal amnesics tend to find partial but not complete deficits in category learning (*Zaki, 2004*), indicating that the hippocampus—though not the sole region involved—makes an important causal contribution.

In the present work, we ask what computational properties of the hippocampus might allow it to contribute to category learning. Using a neural network model of the hippocampus named C-HORSE (Complementary Hippocampal Operations for Representing Statistics and Episodes), we previously demonstrated how the hippocampus might contribute to learning temporal regularities embedded in continuous sequences of stimuli (temporal statistical learning) and to inference over pairwise associations (*Schapiro et al., 2017b*; *Zhou et al., 2023*). We showed that the heterogeneous properties of the two main pathways within the hippocampus may support complementary learning systems, with one pathway specializing in the rapid encoding of individual episodes and another in extracting statistics over time. This division of labor is analogous to the roles of the hippocampus and neocortex in the classic Complementary Learning Systems framework (*McClelland et al., 1995*), and our proposal was thus that a microcosm of this memory systems dynamic plays out within the hippocampus itself.

Category learning is related to temporal statistical learning in requiring information to be integrated across experiences, with the structure of a category discovered across exposure to individual exemplars, but it is also different in important ways. Category learning involves tracking exemplars composed of separate features that can vary in different ways across exemplars. The regularities in these features often manifest in co-occurrence in space at one moment (e.g., different parts of an object), as opposed to co-occurrence nearby in time. There is also often demand in category learning tasks for more explicit grouping and labeling of exemplars. The present work evaluates to what extent the principles of structure learning that allow the hippocampus to support statistical learning may also apply to this different learning domain. If the principles generalize, it would suggest the possibility of broad, domain-general learning mechanisms at work in the hippocampus that allow integration of varied forms of information across experiences.

C-HORSE comes from a lineage of models developed to explain how the subfields of the hippocampus support episodic memory (*Ketz et al., 2013*; *Norman and O'Reilly, 2003*; *O'Reilly and Rudy, 2001*). It instantiates the broad anatomical structure of the hippocampus: hippocampal

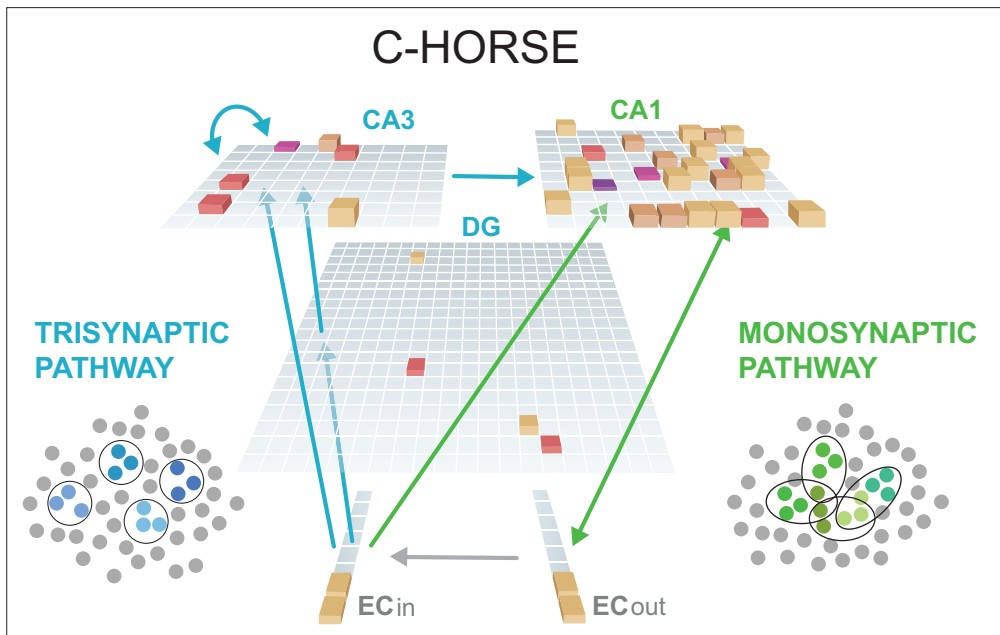

**Figure 1.** C-HORSE architecture. The model consists of dentate gyrus (DG), CA3, and CA1 subfields which map inputs from superficial (EC$_{in}$) to deep layers (EC$_{out}$) of the entorhinal cortex. The height and color of each box represents the activity level of a unit. The trisynaptic pathway (TSP) connects EC to CA1 via DG and CA3 (blue arrows), and the monosynaptic pathway (MSP) connects EC directly with CA1 (green arrows). The TSP specializes in pattern separation (depicted as separated blue pools of neurons), whereas the MSP contains more overlapping representations (overlapping green pools).

subfields dentate gyrus (DG), cornu ammonis (CA3), and CA1 are represented as three hidden layers; they receive input and process output through entorhinal cortex (EC; *Figure 1*). The subfields are connected via two main pathways: the trisynaptic pathway (TSP) and the monosynaptic pathway (MSP). The TSP runs through DG and CA3 to reach CA1. The projections within the TSP are sparse, enabling the formation of orthogonalized representations even with highly similar input patterns (i.e., pattern separation). This corresponds to the observed physiology; for example, distinct sets of place cells in rodents are responsive to particular locations in CA3 even in very similar enclosures (*Leutgeb et al., 2004*). The TSP is highly plastic in the brain and in the model, which supports rapid, even one-shot learning (*Nakashiba et al., 2008*). The TSP also contributes to the process of retrieving previously encoded patterns from partial cues (i.e., pattern completion) via recurrent connections in CA3. The TSP is thus critical for carrying out the episodic memory function of the hippocampus.

The MSP connects EC directly to CA1. These projections do not have the specialized sparsity of those in the TSP, allowing for more overlapping representations to emerge. Place cell responses in CA1 tend to overlap as a function of the similarity of the enclosure (*Leutgeb et al., 2004*). In addition, the MSP seems to learn more slowly (*Lee et al., 2004*; *Nakashiba et al., 2008*). These properties of relatively more overlapping representations and more incremental learning mirror those of neocortex (*McClelland et al., 1995*). In earlier versions of this model (*Norman and O'Reilly, 2003*), the MSP was seen as merely a translator between the TSP representations and neocortex, but we have argued that its properties may make the MSP well suited to learning structured information across episodes (*Schapiro et al., 2017b*). Despite its analogous properties, the MSP is not redundant with neocortex in this framework: the MSP allows *rapid* structure learning, on the timescale of minutes to hours, whereas the neocortex learns more slowly, across days, months, and years. The learning rate in the MSP is intermediate between the TSP (which operates as rapidly as one shot) and neocortex. The proposal is thus that the MSP is crucial to the extent that structure must be learned rapidly.

To investigate the role of the hippocampus in category learning, we tested how the MSP and TSP of C-HORSE contribute to category learning behavior across three different types of categories. First, we evaluated the network's ability to learn simple nonoverlapping categories of exemplars consisting of multiple discrete features, with some features shared among the members of a category and others unique to each exemplar (*Schapiro et al., 2017a*; *Schapiro et al., 2018*). We assessed the model's memory for these different kinds of features as well as its ability to generalize to novel exemplars. Second, we simulated the probabilistic Weather Prediction Task (*Knowlton et al., 1996*; *Knowlton et al., 1994*). In this task, four different cards with shapes are each probabilistically associated with one of two categories: on each trial, a prediction about the weather (sun or rain) is made based on a combination of one, two, or three presented cards. We assessed the model's categorization ability as well as recognition of particular card combinations. Third, we tested the network's ability to learn categories with varying typicality defined along a continuum of overlapping features (*Zeithamova et al., 2008*; *Bowman et al., 2020*). Prototypes of two categories have no features in common, and category exemplars then fall on a continuum between the two prototypes. Exemplars that share more features with the prototype are more typical category members. We assessed the model's categorization and recognition as a function of typicality.

Across the three category learning tasks, C-HORSE was able to both determine the category membership of exemplars and recognize specifics of individual studied exemplars, demonstrating similar learning trajectories to humans in these tasks. There was a division of labor across the two pathways of the hippocampus in these functions: the MSP played a central role in learning the regularities underlying category structure and excelled in generalizing knowledge to novel exemplars. The TSP also contributed to behavior across the tasks but with the opposite expertise, specializing in memory for the unique properties of exemplars. The rapid binding and pattern separation abilities of the TSP that make the pathway well suited to episodic memory are also advantageous for encoding arbitrary relationships in category learning. The findings together motivate a theory of hippocampal contributions to category learning, with the MSP responsible for true understanding of category structure and the TSP for encoding the arbitrary specifics of individual exemplars.

## Methods

We adopted a neural network model of the hippocampus developed after a lineage of models used to explain how the DG, CA3, and CA1 subfields of the hippocampus contribute to episodic memory

(**Ketz et al., 2013**; **Norman and O'Reilly, 2003**; **O'Reilly and Rudy, 2001**). This variant, C-HORSE, was developed recently to account for the role of the hippocampus in statistical learning (**Schapiro et al., 2017b**; **Zhou et al., 2023**). Simulations were performed in the Emergent simulation environment (version 7.0.1, **Aisa et al., 2008**, **O'Reilly et al., 2014a**). Files for running the model can be found on GitHub (copy archived at **Schapiro Lab, 2022**).

## Model architecture

The model has three hidden layers, representing DG, CA3, and CA1 hippocampal subfields, which learn to map input from superficial to deep layers of entorhinal cortex ($EC_{in}$ and $EC_{out}$; **Figure 1**). There is also a separate Input layer (not shown in **Figure 1**) with the same dimensionality as $EC_{in}$, where external input was clamped (i.e., forced to take on particular values), allowing activity in $EC_{in}$ to vary as a function of external input as well as $EC_{out}$ activity. There are one-to-one non-learning connections between Input and $EC_{in}$ and between $EC_{in}$ and $EC_{out}$.

Each layer is composed of a pool of units. There were 400 units in DG, 80 units in CA3, and 100 units in CA1, while input and output layer size (Input, $EC_{in}$, and $EC_{out}$) varied as a function of the task. The hidden layer size ratios reflect the approximate ratios in the human hippocampus (**Ketz et al., 2013**). Units have activity levels ranging from 0 to 1, implementing a rate code. A unit's activity is proportional to the activity of all units connected to it, weighted by connection weights between them. Unit activity is also modulated by inhibition between units within a layer. The inhibition simulates the action of inhibitory interneurons and is implemented using a set-point inhibitory current with k-winner-take-all dynamics (**O'Reilly et al., 2014b**). All simulations involved tasks with discrete-valued dimensions as these are more easily amenable to implementation across input/output units whose activity tends to become binarized as a result of these inhibition dynamics. It will be important for future work to extend to implementations of category learning tasks with continuous-valued dimensions.

The TSP connects EC to CA1 via DG and CA3. Connections are sparse, reflecting known physiological properties of the hippocampus: DG and CA3 units receive input from 25% of units in the $EC_{in}$ layer, and CA3 receives directly from 5% of DG. Both DG and CA3 have high levels of within-layer inhibition. CA3 also has a full recurrent projection (every unit connected to every other). Finally, CA3 is fully connected to CA1.

The MSP is formed by a full direct connection from $EC_{in}$ to CA1, and bidirectional connections between CA1 and $EC_{out}$. CA1 has lower inhibition than CA3 and DG, allowing a higher proportion of units in the layer to be simultaneously active. See **Supplementary file 1g and h** for parameter values.

## Learning

The model was trained as an autoencoder, adjusting connection weights to reproduce patterns presented to $EC_{in}$ on $EC_{out}$. Weights were updated via Contrastive Hebbian Learning (**Ketz et al., 2013**). Following **Ketz et al., 2013** and in line with known changes in projection strength across the theta phase, each learning trial has two minus phases each contrasted against one plus phase, with the minus phases corresponding to the peak and trough of the theta phase. At the simulated trough, EC has a strong influence on CA1, while the connection from CA3 to CA1 is inhibited. At the simulated peak, CA3 has a strong influence on CA1, while connections from $EC_{in}$ to CA1 are inhibited. This learning scheme allows the two pathways to learn more independently. During the plus phase, the target output is directly clamped on $EC_{out}$. Weights are adjusted after each trial to reduce the local differences in unit coactivities between each of the two minus phases and the plus phase. As in previous work, the learning rate on the TSP was set to be 10 times higher than the MSP (**Ketz et al., 2013**; **Schapiro et al., 2017b**), at 0.02 versus 0.002, except in the third simulation, where the MSP learning rate was smaller to accommodate stimuli with high degrees of overlap (which can lead to degenerate learning at relatively higher learning rates). The differential learning rates support the different functions of the TSP and MSP in one shot versus more incremental learning across experiences.

Simulations had a fixed number of training trials except in the Weather Prediction Task, where we used a stopping rule: after a minimum of 25 training trials, the model had to achieve five consecutive trials with sum squared error below 1.2. The stopping rule was introduced because of the probabilistic nature of the categories, where it is not possible to eliminate all error.

## Testing

Connection weights were not changed during test. Networks were tested before any training (trial 0) and intermittently after fixed sets of training trials. For each set of simulations, we assessed the model's categorization ability and its ability to remember item-specific information.

## Lesions

We simulated lesions of the MSP and TSP in order to assess the contributions of each pathway to performance. The MSP lesion was performed by setting the strength of the projection from $EC_{in}$ to CA1 to 0. We did not lesion connections between CA1 and $EC_{out}$ because they are necessary for producing output. The TSP lesion was performed by setting weights from CA3 to CA1 to 0. The lesions were implemented in a version of the model that did not use the theta-inspired learning scheme described above as only one minus phase is appropriate with only one pathway intact. Lesions were implemented during both learning and testing.

## Statistical analysis

For each simulation, we analyzed 100 networks with randomly initialized weights (randomizing both the topography of weight connections where projections were sparse as well as the values of individual weights). To characterize variability across initializations, we conducted statistical analyses treating network initialization as the random effects factor. To compare performance in the intact and lesioned networks, the mean accuracy was submitted to an ANOVA with a between-network factor *Condition* (intact, MSP-only, and TSP-only network), a within-network factor *Trial* (number of training trials prior to test), and *Network initialization* as a random effects factor. Following the omnibus ANOVA, to determine which conditions may differ, we ran three separate ANOVAs with a between-network factor *Condition* which included two out of three conditions (intact vs. MSP-only, intact vs. TSP-only, and MSP-only vs. TSP-only). Data visualization and statistical analyses were performed in R, version 3.6.1 (*R Development Core Team, 2019*).

## Representational similarity analyses

To assess the nature of learned representations in the networks, we performed representational similarity analyses for each of the simulations during a test phase at the end of training. We used Pearson correlation to relate the patterns of activity evoked by presentation of different items. We analyzed representations separately in the 'initial' and 'settled' response to each item in the intact network. The initial response captures the activation pattern once activity has spread throughout the network but before output activity in $EC_{out}$ recirculates back to the input in $EC_{in}$—before there is an impact of 'big-loop' recurrence (*Kumaran and McClelland, 2012*; *Schapiro et al., 2017b*). The settled response captures the fully settled pattern of activity including the influence of big-loop recurrence. Big-loop recurrence permits the representations in CA1 to influence those in DG and CA3, so separate analysis of the initial response allows cleaner assessment of the unique representational contributions of the different subfields.

# Results

## Simulation 1: Learning distinct categories of items with unique and shared features

First, we examined C-HORSE's ability to learn categories of items that consist of multiple discrete features, with some features unique to individual items and others shared amongst members of the same category, and no features overlapping across categories. To test the network's ability to learn these categories, we presented a set of novel objects representing three categories of 'satellites,' with five satellites in each category, following empirical work with this paradigm (*Schapiro et al., 2017a*; *Schapiro et al., 2018*). The model and humans were given a comparable number of training trials: 140 for the model and on average 122 for humans (*Schapiro et al., 2018*).

Each category had a prototype, defining the shared features for that category. Four other exemplars of the category had one out of five features swapped away from the prototype, such that they had one unique feature and four shared features (*Figure 2a*). This structure means that one of the shared features is present across all exemplars in a category, which effectively serves as a category

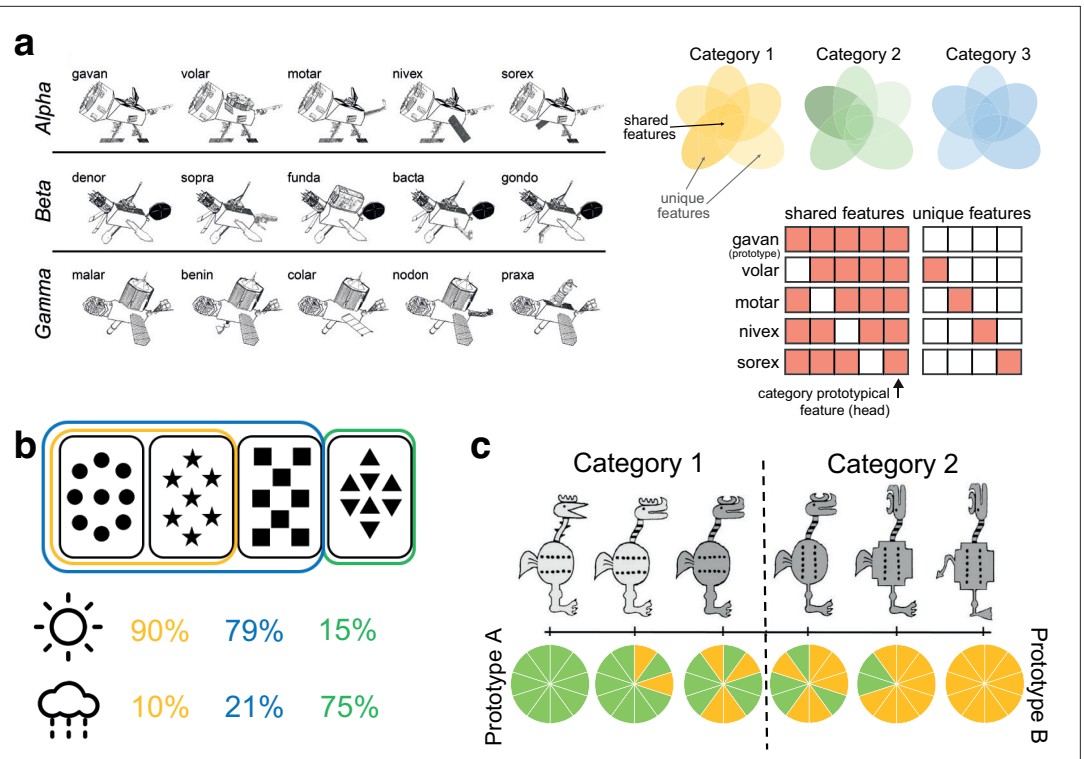

**Figure 2.** Overview of simulated category learning paradigms. (**a**) Satellite categories: distinct categories of novel 'satellites' consisting of unique and shared features (reproduced from Figure 1A of *Schapiro et al., 2017a*). Grids depict model input structure for the five members of the Alpha category, with the prototype consisting of only shared features and all other exemplars containing one unique feature. One feature (the satellite head) is category prototypical, appearing in all exemplars of its category. (**b**) Weather Prediction Task: each abstract card is probabilistically related to a category (sun or rain), and on a given trial, category must be guessed from a simultaneously presented set of 1–3 cards (*Knowlton et al., 1994*). The illustration shows the first two cards related to the 'sun' category on 90% of the trials (and to 'rain' on 10% of the trials), while a combination of the first three cards related to sun 79% of the time, and a fourth card viewed by itself associated with sun 15% of the time. (**c**) Intermixed categories with varying typicality: categories where each item consists of 10 binary features. The two prototypes on opposite sides of the feature space have no features in common (depicted by all green versus all yellow features in the piecharts), and the rest of the exemplars have a varying number of features in common with the prototypes (adapted from Figure 1 of *Zeithamova et al., 2008*).

name/indicator and was used to assess categorization ability. Each feature was assigned to one unit in the input layer. If the feature was present, the input unit representing the feature took on a value of 1, and otherwise 0. Thus, there were 27 input units in total, 9 per category. Within each category, there were five units representing shared features and four units for unique features (*Supplementary file 1a*; see *Figure 2a* for an illustration of the inputs corresponding to one category).

To characterize the network's behavior, we investigated its ability to recognize unique features of individual trained satellites (unique feature recognition), recognize the prototypical feature shared across all trained members of a category (categorization), and fill in the prototypical feature for novel satellites not presented during training (generalization). For unique feature recognition, we presented the network with the unique feature of a trained satellite as input and evaluated the network's ability to activate that feature on the output, compared to unique features of other members of the same category (*Supplementary file 1b*). Accuracy was determined by dividing the activation of the correct unit by the total activation in the four units representing unique features for that category (chance accuracy 0.25).

*Figure 3a and b* show human performance across training for unique features (ability to fill in missing 'code names') and categorization (ability to fill in the category-prototypical visual feature, analogous to model assessment), re-analyzed from first session data from *Schapiro et al., 2017a*. The intact network learned to recognize unique features of individual satellites with a similar trajectory to

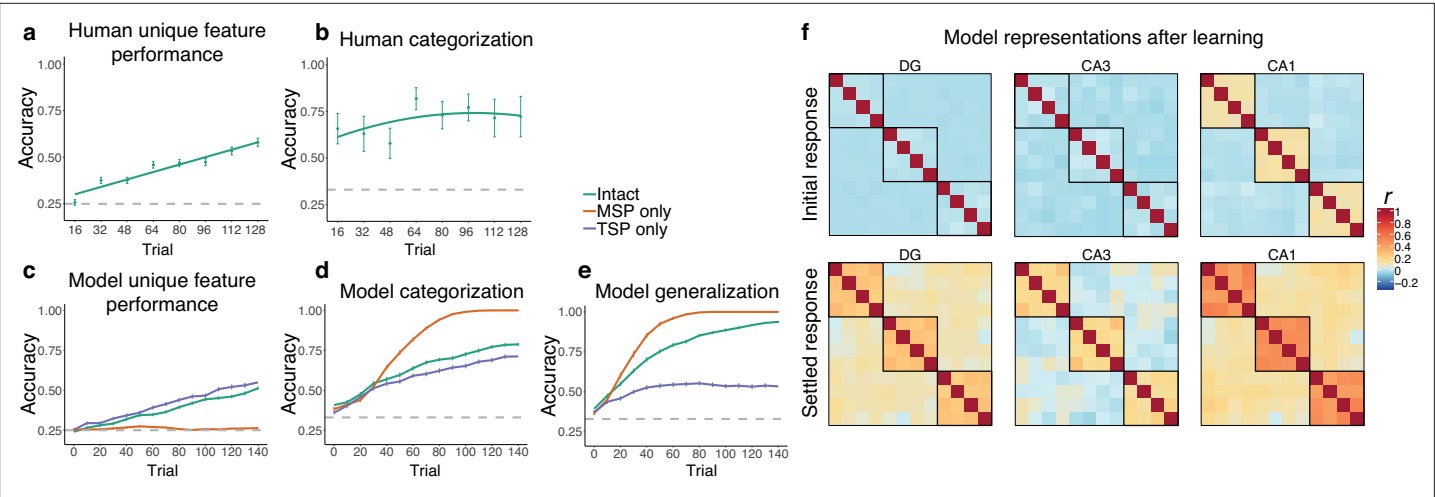

**Figure 3.** Simulation 1: satellite task. Human performance on (**a**) unique features and (**b**) categorization across training in *Schapiro et al., 2017a*. Unique feature trajectory plotted with a linear learning curve fit and categorization with a quadratic fit. (Humans were not tested on generalization over the course of training.) Performance of the network across training trials for (**c**) unique feature recognition, (**d**) categorization of trained items, and (**e**) generalization (categorization of novel items). Performance is shown for the intact network (green), a version of the network with only the monosynaptic pathway (MSP) (orange), and a version with only the trisynaptic pathway (TSP) (purple). Plots show mean performance averaged across random network initializations. Error bars denote ±1 s.e.m. across people / 100 network initializations (some are too small to be visible). Dashed lines indicate chance level performance. Source data can be found in *Figure 3—source data 1*. (**f**) Representational similarity for the initial and settled response of the intact network. Each item appears in the rows and columns of the heatmaps. The diagonals are always 1 as this reflects items correlated to themselves, and the off-diagonals are symmetric. Black boxes delineate categories. Source data can be found in *Figure 3—source data 2*.

The online version of this article includes the following source data for figure 3:

**Source code 1.** R code used to generate *Figure 3* panels.

**Source data 1.** Data corresponding to all line plots in *Figure 3*.

**Source data 2.** Correlation values for representational similarity analysis heatmaps in *Figure 3f*.

humans (*Figure 3c*). A version of the network with access only to the MSP was completely unable to output the correct unique features, whereas a version with only the TSP could do this well above a chance, and even slightly better than the intact model. This reveals that the TSP is fully responsible for the network's ability to remember unique features. Differences between model types across time were all highly reliable (all $p_s < 0.001$).

To test categorization ability, we examined the network's ability to indicate the correct category of a satellite, operationalized as activating the category-prototypical feature shared across all members of the satellite's category. We presented the network with the unique feature of a satellite and divided output activity for the correct prototypical feature by the sum of activation of the three prototypical features for the three categories. The model's trajectory and level of performance (*Figure 3d*) were comparable to humans (*Figure 3b*), although humans were able to perform this task well very early on in training. The MSP-only network exhibited much better performance than the intact network. The TSP-only network had poorer performance (0.71 vs. 1 by the end of training), but still well above chance. Because this test involved trained satellites, categorization could be solved using a memorization strategy, likely enabling good performance for the TSP-only network. Because the MSP was unable to remember unique features (*Figure 3c*), it expressed knowledge only of the shared features, leading to excellent categorization performance. The intact network combined information from both sources, resulting in intermediate performance (see below for discussion of the idea that a control mechanism might allow selective enhancement of pathways depending on task). All differences between model types were again highly reliable (all $p_s < 0.001$).

The strongest test of category understanding is the ability to generalize to novel instances. To test generalization, we presented the network with a set of 18 satellites (6 per category) that were not presented during training (*Supplementary file 1c*). Each input satellite consisted of two shared features (not including the category-prototypical feature) and two unique features, and we tested the network's ability to output the category-prototypical feature. A similar pattern was observed as with

the categorization of familiar items, but the TSP-only network showed poorer (though still above-chance) performance (0.53) in comparison to the intact network (0.94), while the MSP-only network was even better (*Figure 3e*). The MSP was able to ignore the unique features of these novel satellites, resulting in perfect generalization behavior relatively early in training. All differences were reliable ($p_s < 0.001$).

To assess network representations, we performed representational similarity analysis for each hidden layer of the intact network at the end of training (140 trials). We captured the patterns of unit activities evoked by presentation of each satellite's unique feature (for the 12 satellites with unique features). There was no structure in the representations prior to training (not depicted), and the representations that emerged with training revealed sensitivity to the category structure (*Figure 3f*). This was particularly evident in CA1, with items from the same category represented much more similarly than items from different categories. We separately analyzed the initial pattern of activity evoked by each feature (before there was time for activity to spread from $EC_{out}$ to $EC_{in}$), and the fully settled response. The initial response allows us to understand the separate representational contributions of the subfields, before CA1 activity has the potential to influence DG and CA3. In the initial response, there was no sensitivity to category structure in DG and CA3—items were represented with distinct sets of units. This is a demonstration of the classic pattern separation function of the TSP, applied to this domain of category learning, where it is able to take overlapping inputs and project them to separate populations of units in DG and CA3. CA1 representations, on the other hand, mirrored the category structure, with overlapping sets of units evoked by items in the same category. This result is consistent with our neuroimaging findings using this paradigm, where CA1 was the only subfield of the hippocampus to show significant within versus between category multivoxel pattern similarity (*Schapiro et al., 2018*). The settled response revealed sensitivity to category structure in all three hidden layers, reflecting the influence of CA1 on the rest of the network after 'big-loop' recurrence (*Koster et al., 2018*; *Kumaran and McClelland, 2012*; *Schapiro et al., 2017b*), though CA1 still showed the strongest response.

In sum, these results suggest that the network is capable of learning categories and generalizing to novel instances. To achieve this, the MSP and TSP take on complementary roles: the MSP extracts regularities and learns information that defines category structure, while the TSP encodes individual exemplars and handles unique feature recognition. These properties are analogous to our prior simulations with C-HORSE, where we found that the MSP detects statistical structure while the TSP encodes episode-unique information (*Schapiro et al., 2017b*). There are two key properties that differ between the pathways that underlie these results: (1) slower learning in the MSP than TSP, which allows integration of information over longer periods of time in the MSP and rapid learning in the TSP, and (2) more overlapping (distributed) representations in the MSP than TSP, which helps the MSP see commonalities across experiences and helps the TSP separate experiences to avoid interference.

## Simulation 2: Learning probabilistic categories

While the previous set of simulations focused on deterministic categories, that is, categories in which each item could belong to only one category, in this section we test the network's ability to learn probabilistic categories using the canonical Weather Prediction Task (*Knowlton et al., 1996*; *Knowlton et al., 1994*; *Reber et al., 1996*). In this task, there are a total of four cards with abstract shapes, and a combination of one, two, or three cards is simultaneously presented on each trial to predict a weather outcome, sunshine or rain (*Figure 2b*).

The number of times a particular combination of cards was presented to the network and the frequency of its association to each category were identical to the experimental procedure used in *Knowlton et al., 1994* (*Supplementary file 1d*). Two combinations of cards that had an equal probability of being associated with each category were removed from analysis. Each card was represented by one unit in the input and output, and each weather outcome (category) was represented by two units (increasing the relative salience of category information). As in prior simulations, the model was trained as an autoencoder, meaning that both the cards and category information were presented as input and the model was asked to reconstruct all of these features on the output layer. This training regimen is more akin to an 'observational' than 'feedback' mode of the task, which is appropriate given evidence that the medial temporal lobe (MTL) is more engaged by observational variants (*Poldrack et al., 2001*; *Shohamy et al., 2004*). We trained the network for

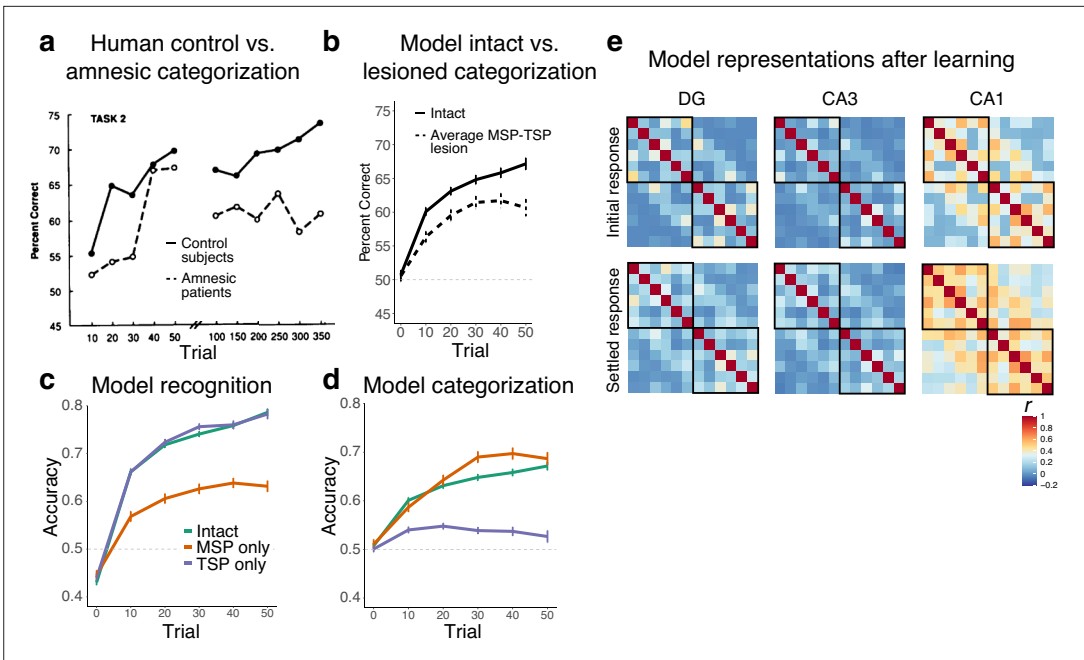

**Figure 4.** Simulation 2: Weather Prediction Task. (**a**) Human control and amnesic performance from Task 2 of *Knowlton et al., 1994* (adapted from Figure 2). (**b**) Intact and lesioned model categorization performance across trials, simulating the initial phase of human learning. (**c**) Model recognition performance for the intact network, a version of the network with only the monosynaptic pathway (MSP), and a version with only the trisynaptic pathway (TSP). (**d**) Model categorization performance across the three network variants. Source data can be found in *Figure 4—source data 1*. (**e**) Representational similarity for the initial and settled response of the intact network. Each combination of cards appears in the rows and columns of the heatmap, organized by most likely to predict sun to most likely to predict rain. Source data can be found in *Figure 4—source data 2*.

The online version of this article includes the following source data for figure 4:

**Source code 1.** R code used to generate *Figure 4* panels.

**Source data 1.** Data corresponding to all line plots in *Figure 4*.

**Source data 2.** Correlation values for representational similarity analysis heatmaps in *Figure 4e*.

50 trials, simulating the first part of Task 2 in *Knowlton et al., 1994*, where patients also saw 50 trials (*Figure 4a*).

To examine the network's performance, we tested its ability to reconstruct individual combinations of cards (recognition) and predict category based on the presented cards (categorization). For recognition performance, we evaluated reconstruction of the correct card output units given a set of input cards. Since all possible card combinations are presented during training, this does not involve discriminating old from new combinations, but is rather a simple measure of the network's ability to process each distinct card configuration. Recognition score was calculated by dividing the mean activation of correct card units by the mean activation across all card units.

A stopping criterion was used during training, resulting in networks being trained for different numbers of trials, with better performing networks stopping earlier (which sometimes produces a slight dip in performance towards the end of training, as seen in *Figure 4*). We ran as many networks as needed to obtain data from 100 networks in each of the three lesion conditions at trial 50. As a result, there were 722 networks at trial 10, 715 networks at trial 20, 625 networks at trial 30, 443 networks at trial 40, and 300 networks in the final test trial (100 per condition).

The results indicated that the network was able to recognize individual combinations of cards (*Figure 4c*). An ANOVA revealed significant main effects of trial, lesion type, and their interaction (all $p_s < 0.001$). While the intact and TSP-only network showed equivalent performance (p=0.138), both showed significantly higher recognition accuracy than the MSP-only network ($p_s < 0.001$). Consistent with the prior simulations, the TSP-only network demonstrated better recognition than the MSP-only

network, and performed in this case virtually identically to the intact network. For this simple form of recognition, the MSP-only network was able to perform above chance.

Categorization performance was assessed by presenting sets of cards without any category input and testing the network's ability to output the correct category. The intact and the MSP-only networks were able to categorize the sets of cards more effectively than the TSP-only network (*Figure 4d*). The trajectory and accuracy levels for the intact and MSP-only network were similar to the performance levels observed in healthy participants (*Figure 4a*). The TSP-only network performed close to chance on this task (though still exhibiting some degree of reliable categorization ability). An ANOVA revealed significant main effects of trial, lesion type, and their interaction (all $p_s<0.001$). Further analyses revealed significant differences between all three lesion conditions (intact vs. MSP-only network: $p=0.005$, other $p_s<0.001$). As in categorization and generalization in Simulation 1, the network performed better in categorization without the presence of the TSP, though this advantage was more subtle in this simulation. *Figure 4b* shows categorization for the intact network replotted as well as an average of the behavior of the MSP and TSP lesioned networks as a simulation of the behavior of amnesic patients shown in *Figure 4a* (who had a range of types and degrees of hippocampal damage).

*Figure 4e* shows the similarity structure of the activity patterns evoked by different card combinations at the end of learning. As in Simulation 1, DG and CA3 represented the card combinations relatively distinctly, whereas the patterns of activity in CA1 reflected the category structure (mean similarity within category: 0.38, across: 0.25; $p<0.001$). Similarity levels increased somewhat throughout the network after there was opportunity for big-loop recurrence (the 'settled' response), but patterns remained quite distinct in the TSP. The probabilistic nature of the categories in this simulation resulted in much noisier representations as well as middling behavioral performance in the model, but the qualitative match to human behavior (*Figure 4a* vs. *Figure 4b*) suggests that these underlying representations may be related to those supporting human behavior.

In sum, the network learned probabilistic categories similarly to humans. Mirroring the Simulation 1 results, the MSP contributed relatively more to this categorization ability, whereas the TSP was better able to process individual combinations of cards.

## Simulation 3: Learning intermixed categories with varying typicality

The third set of simulations tested the network's ability to acquire categories with intermixed features and varying typicality. The network was exposed to a set of novel creatures belonging to two categories (*Figure 2c*; *Zeithamova et al., 2008*). Each creature had 10 binary features, and prototypes of the two categories had no features in common. The rest of the items spanned a continuum between the two prototypes: some items had nine features in common with one prototype and one feature in common with the other prototype; other items had eight features shared with one prototype and two features shared with the other prototype, and so on. If an item had more than five features in common with one prototype, it was considered to belong to the prototype's category (*Figure 2c*). Each feature was represented by two units (one unit for each of the two possible feature values), and each category label was represented by five units (increasing the salience of category information relative to the many creature features).

During training, the network learned 20 items, 10 from each category. The model saw each item five times for a total of 100 trials. The stimulus structure and training were based on *Zeithamova et al., 2008*, where participants were presented with 4 runs of 20 items, 10 from each category. Within each category, there were two items that shared nine features with the prototype, three items with eight shared features, three items with seven shared features, and two items with six shared features (*Supplementary file 1e*). At test, the network was presented with the training set and a test set consisting of 42 novel items (*Supplementary file 1f*): the two untrained prototypes and five items at each distance from the prototype (*Zeithamova et al., 2008*). We tested the network's ability to remember the atypical features of the training items (atypical feature recognition) and its ability to predict the correct category for the novel items (generalization).

Human generalization performance in this paradigm across learning and in a subsequent test from *Bowman et al., 2020* is shown in *Figure 5a and b*. Participants improved for all item types across learning, with better performance for more prototypical items. Generalization was assessed in the model by testing the network's ability to predict the correct category for a set of novel category exemplars based on their features only (no category information was inputted). The mean activation in

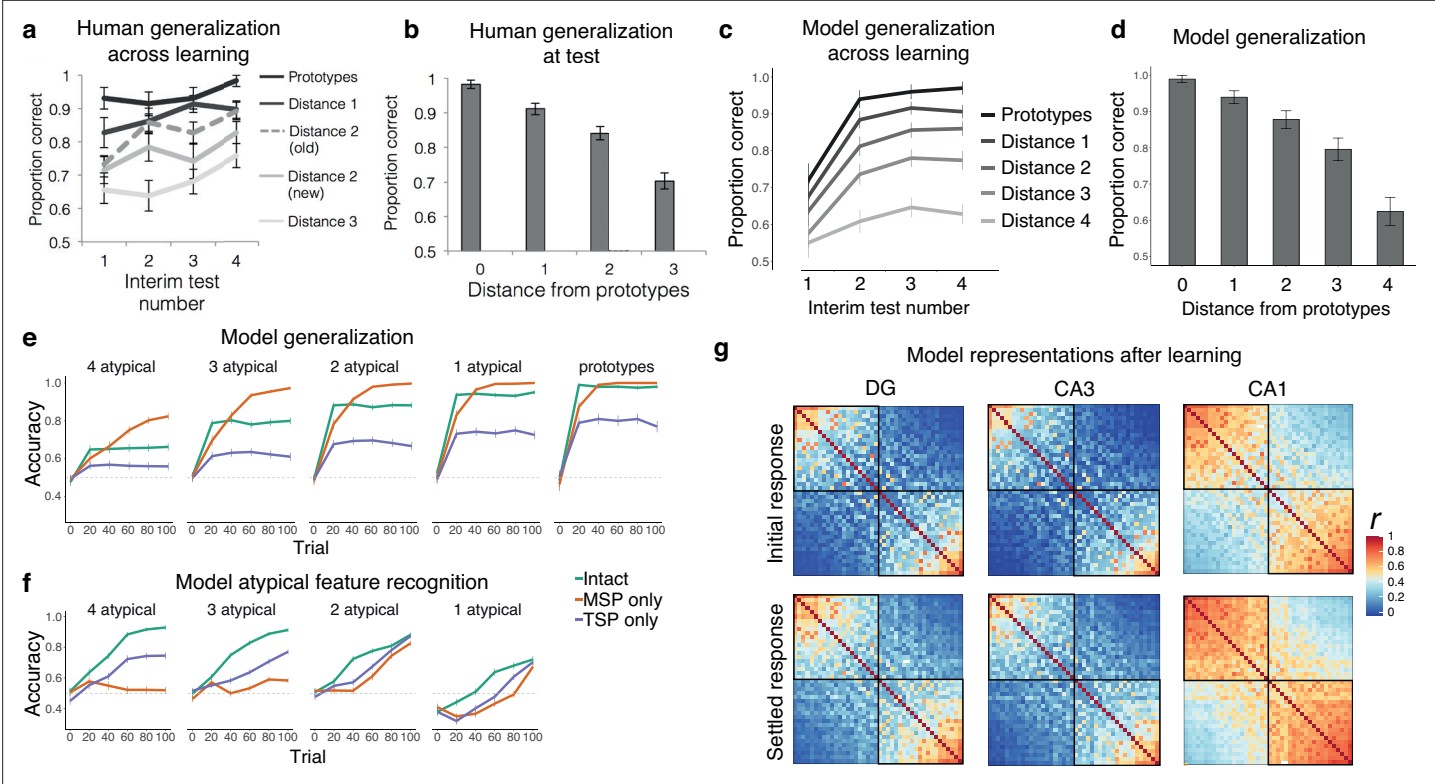

**Figure 5.** Simulation 3: intermixed categories with varying typicality. Human generalization across (**a**) learning and (**b**) test, for varying levels of typicality, from Figure 3 in *Bowman et al., 2020*. Intact model generalization across (**c**) learning, with 10 trials prior to each interim test, and (**d**) at the end of learning. Source data can be found in *Figure 5—source data 1*. (**e**) Model generalization broken down by typicality and model type. (**f**) Model atypical feature recognition broken down by typicality and model type. Source data can be found in *Figure 5—source data 2*. (**g**) Representational similarity for the initial and settled response of the intact network. Each item appears in the rows and columns of the heatmap, organized by most prototypical members of one category to most prototypical members of the other. Source data can be found in *Figure 5—source data 3*.

The online version of this article includes the following source data for figure 5:

**Source code 1.** R code used to generate *Figure 5* panels.

**Source data 1.** Data corresponding to *Figure 5c, d*.

**Source data 2.** Data corresponding to *Figure 5e, f*.

**Source data 3.** Correlation values for representational similarity analysis heatmaps in *Figure 5g*.

units representing the correct category was divided by the mean activation across units representing both the correct and incorrect categories. Like human participants, the intact network improved in its categorization of novel items across training, with better performance for more prototypical items (*Figure 5c*). *Figure 5c* plots the initial training period, with 10 trials prior to each interim test, *Figure 5d* shows the typicality gradient at the end of this training, and *Figure 5e* shows generalization behavior over a longer training period, broken down by pathway. As in the prior simulations, the MSP-only network excelled in generalization, outperforming the intact network by the end of training across all typicality levels. The TSP-only network performed much worse, but still above chance, and was able to generalize quite well for highly prototypical items. All main effects and interactions were significant ($p_s < 0.001$).

Atypical feature recognition was assessed by testing the ability to activate the correct atypical features in the output layer when presented with trained category exemplars. For each item, we compared the activation of features that did not match the prototypical item (atypical features) to the total activation in the atypical units. The proportion of activation in the correct atypical features was compared against chance (0.1 for items that had only one atypical feature, 0.2 for items that had two atypical features, etc.). As shown in *Figure 5f*, the intact network generally showed good atypical feature recognition performance. The level of recognition accuracy depended on the level of similarity

of the item to its prototype. Atypical features of less typical category members were recognized more easily than atypical features for items very similar to the prototype. Unlike prior simulations, in this case the intact network exhibited better performance than the lesioned networks, suggesting that this task benefits from having both pathways intact. The TSP-only network performed better than the MSP-only network, which was virtually unable to recognize atypical category members (three or four atypical features), but showed somewhat better performance on items more similar to the prototype (one or two atypical features). The MSP can thus contribute to atypical feature memory to some extent, when the item is overall very similar to the prototype. The more arbitrary the item, the more the TSP is needed. Even for arbitrary items, though, the TSP benefited from the presence of the MSP (as indicated by higher performance for the intact network), suggesting that this may be a situation where the MSP plays an important supportive function. Initial below-chance performance for the exemplars with only one atypical feature reflects the tendency to pattern-complete these items to the highly similar prototype. Effects of lesion, time, number of shared features, and their interactions were all significant ($p_s<0.001$).

Visualization of the internal representations of the model after training provides insight into the behaviors of the two pathways. *Figure 5g* shows the similarity of the evoked activity of the items in this domain, with items arranged from most prototypical members of one category to most proto-typical members of the other. As in the prior simulations, DG and CA3 represented the items more distinctly than CA1, and settled activity after big-loop recurrence increased similarity, especially in CA1. This simulation was unique, however, in that DG and CA3 showed clear similarity structure for the prototype and highly prototypical items. There is a limit to the pattern separation abilities of the TSP, and these highly similar items exceeded that limit. This explains why, at high typicality levels, the TSP could be quite successful on its own in generalization (*Figure 5e*), and why it struggled with atypical feature recognition for these items (*Figure 5f*).

Overall, the results from Simulation 3 are convergent with those from Simulations 1 and 2, with the TSP contributing more to recognition than categorization and the MSP contributing more to catego-rization than recognition. As in the satellite simulation, there was a clear trade-off across pathways in categorization behavior, with the MSP-only network performing better without the influence of the TSP. The recognition results showed an interesting new dimension of behavior as a function of exem-plar typicality: the TSP is better than the MSP at remembering the unique features of more atypical exemplars. The more features an exemplar has that depart from the category prototype, the more important the arbitrary binding ability of the TSP.

## Discussion

We found that a neural network model of the hippocampus was readily able to learn three different types of categories, providing an account of how the hippocampus may contribute to category learning. Across paradigms, the MSP specialized in detecting the regularities that define category structure. Lower sparsity in this pathway enables overlapping, distributed representations (*Hinton, 1984*), which facilitate the detection of commonalities across exemplars, and a relatively lower learning rate helps to integrate this information over time. After learning, representations of items from the same category were more similar than items from different categories. This was driven by and especially true in subfield CA1, consistent with our prior fMRI finding that CA1 shows stronger within-category representational similarity (*Schapiro et al., 2018*). The work thus demonstrates that the principles that allowed C-HORSE to detect regularities in structured temporal input (*Schapiro et al., 2017b*) apply more broadly to detecting regularities in multidimensional category spaces. The fact that one model accounts for findings across these paradigms points to domain-general principles in the operations the hippocampus is engaged in to extract related information across experiences.

In contrast to the MSP's capacity for detecting shared structure and generalizing, the main contri-bution of the TSP to category learning was encoding the distinguishing information about individual category exemplars. Higher sparsity in this pathway allowed the TSP to orthogonalize similar inputs and encode the details of individual exemplars. The ability to quickly bind together arbitrary information that is so useful for episodic memory (e.g., *Norman and O'Reilly, 2003*) translates into a specialization for remembering the arbitrary details of individual exemplars in the domain of category learning. This ability proved especially useful for atypical exemplars. Lesioning the TSP resulted in poor recognition with preserved categorization ability. Consistent with these behaviors, a recent study found that TSP

white matter integrity predicts the ability to learn category exceptions (*Schlichting et al., 2021*). We thus propose that the properties of the TSP should make it useful beyond its traditional domain of episodic memory—it should contribute to any rapid new learning that requires memory for arbitrary, as opposed to systematic, information.

## Relationship to other models of categorization

Our goal was to take a model with an architecture inspired by the anatomy and properties of the hippocampus and explore how it might accomplish category learning. While we did not endeavor to build in any particular strategies, the emergent behaviors of the model bear resemblance to existing psychological models of categorization. The model thus provides a bridge across levels of analysis, showing how neurobiological mechanisms may give rise to some of the more abstract operations of existing models.

The classic exemplar model proposes that people store memory representations of individual category instances and perform similarity judgments on these separate representations at test in order to come to a categorization decision (*Medin and Schaffer, 1978*; *Nosofsky, 2011*; *Nosofsky and Johansen, 2000*). This model has been successful in accounting for many findings across categorization and recognition paradigms (*Nosofsky, 1988*; *Nosofsky, 1991*; *Nosofsky and Zaki, 1998*; *Palmeri, 1997*). The TSP of our model is similar to the exemplar model in that it stores separate traces of individual exemplars. In fact, our model provides an account of how a neural circuit might implement exemplar-style representations: the sparsity-inducing machinery that leads to pattern separation of individual episodic memories in the TSP similarly leads to pattern separation across exemplars. The consequence in our model is high fidelity memory for the details of particular exemplars. Unlike the exemplar model, however, our model's TSP exhibited relatively poor categorization. There may be modifications to the model that would allow the TSP to behave more like an exemplar model. For example, the present version of the model does not modulate the influence of the DG during encoding and retrieval, but it is possible that reducing the influence of DG during retrieval would bias the TSP toward pattern completion at test (*Lee and Kesner, 2004*; *Rolls, 1995*; *Rolls, 2018*), which might enhance certain kinds of categorization. REMERGE (*Kumaran and McClelland, 2012*) is a model of how the hippocampus might support inference and generalization that relies on pattern separated, conjunctive representations, as in our TSP. The model can accomplish categorization in a manner closely analogous to exemplar models (*Kumaran and McClelland, 2012*, Appendix), suggesting that there may indeed be ways to increase the categorization ability of a TSP-style representation. Regardless, and across these models, the unique expertise of the TSP-style representation is in its ability to retain the details of individual exemplars.

The classic prototype model postulates that categories are represented by the central tendency across exemplars in a category, without retaining traces of the individual observed exemplars (*Minda and Smith, 2011*). The prototype model explains categorization behavior well in the context of well-defined, high-coherence categories (*Bowman and Zeithamova, 2020*; *Minda and Smith, 2001*). The MSP of our model behaves similarly to a prototype model in that it tends to abstract across the details of individual exemplars and represent the central tendency. However, this is not true in an absolute sense—the representation in the MSP is sensitive to individual exemplars to some extent.

*McClelland and Rumelhart, 1985* showed how specifics and generalities can coexist in a neural network model with distributed representations. Our MSP uses distributed representations and shows some degree of this dual sensitivity. However, there is a tension between the representation of specifics and generalities in the way that the hidden layers in our model behave. In a hidden layer with very large capacity and a very slow learning rate, distributed internal representations can be carefully and gradually shaped to faithfully reflect the statistics of the environment, which can include representation of both arbitrary and systematic information, to the extent that each is present in the input. Neocortical areas of the brain likely have this property of representing arbitrary and general information in harmonious superposition, as in the representations described by *McClelland and Rumelhart, 1985*. But in the case of our hippocampal system, capacity is somewhat more limited and, critically, learning rates are necessarily fast, in order to support behavior on the timescale of a few minutes to hours. The fast learning rate appears to force trade-offs: representations can either tend to emphasize the specifics or the generalities.

Our model assumes that every item is encoded in two different ways, with one representation focusing on its details, separating it from other similar items, and the other glossing over the details, emphasizing its similarity to other items. This idea is consistent with neuroimaging data showing coexisting neural representations that are more prototype- and exemplar-like (*Bowman et al., 2020*). This perspective avoids the kind of discrete category decision-making that occurs in a category learning model like SUSTAIN, where a new exemplar either merges with an existing category or separates into a new one (*Love et al., 2004*). We propose that the brain may have it both ways, solving the tension between representing details and generalities by maintaining both representations in parallel. The solution is closely analogous to that proposed by the Complementary Learning Systems theory, which argued that the hippocampus and neocortex take on complementary roles in memory for encoding the specifics of new items and generalizing across them over time (*McClelland et al., 1995*). The MSP in our model has properties similar to the neocortex in that framework, with relatively more overlapping representations and a relatively slower learning rate, allowing it to behave as a miniature semantic memory system. The TSP and MSP in our model are thus a microcosm of the broader Complementary Learning Systems dynamic, with the MSP playing the role of a *rapid* learner of novel semantics, relative to the slower learning of neocortex.

There are many models of category learning in the literature, including neural network models, that would likely exhibit behavior closely analogous to our model's MSP. Our goal here is not to claim that our model better fits empirical data than existing models, but rather to provide a proof-of-concept demonstration of how the computations of hippocampal subregions may give rise to different components of category learning. Detailed comparison of the model's behavior to other models in the literature will be valuable in evaluating and refining the model, however, and will be an important goal for future work.

## Coordinating the contributions of the MSP and TSP

Having two different representations of the same item leads to a problem at retrieval: which representation should be used? In our current work, we have assumed that both representations contribute, and the retrieved information reflects some blending of the two. But in many cases, there is a trade-off in the relevance and utility of the representations, depending on the task. Such trade-offs between representing specifics and regularities have been documented in the literature (e.g., *Sherman and Turk-Browne, 2020*). We found several cases of trade-offs playing out in our simulations; for example, generalization in the satellite categories is strong in the intact model, which uses both pathways, but much stronger in the version of the model that only uses the MSP. This suggests that a control mechanism that enhances one pathway over another depending on the task would be beneficial for behavior. In a recent paper, we adopted a version of C-HORSE that invoked such a control function in order to explain behavior across tasks with different demands in an associative inference paradigm (*Zhou et al., 2023*). Medial prefrontal cortex could potentially carry out a control function of this kind (*Sherman et al., 2023*), as it participates in category learning (*Mack et al., 2020*) and is known to modulate CA1 representations as a function of task (*Eichenbaum, 2017*; *Guise and Shapiro, 2017*). As the TSP and MSP are both routed through CA1, medial prefrontal control over CA1 could conceivably help coordinate information flow there for optimal behavior. This will be an interesting hypothesis to explore in future modeling and empirical work.

## Hippocampal maturation and development of categorization abilities

In humans, the hippocampus has a protracted development, with hippocampal subfields exhibiting different maturation rates (*Lavenex and Banta Lavenex, 2013*). While the CA1 subfield develops during the first two years of life and reaches adult-like volume around two years, the DG and CA3 subfields develop at a slower pace (*Bachevalier, 2013*; *Gómez and Edgin, 2016*; *Lavenex and Banta Lavenex, 2013*). The projection from EC to CA1 (the MSP), develops prior to the projection from EC to DG in the TSP (*Hevner and Kinney, 1996*; *Jabès et al., 2011*). Given the MSP's role in detecting regularities, early maturation of CA1 suggests that the ability to detect regularities should emerge early in development. Indeed, even before their first birthday infants show evidence of categorization (*Eimas and Quinn, 1994*; *Mareschal and Quinn, 2001*; *Younger and Cohen, 1983*) and statistical learning abilities (*Fiser and Aslin, 2002*; *Kirkham et al., 2002*; *Saffran et al., 1996*). There is evidence for involvement of the anterior hippocampus in statistical learning as young as three months (*Ellis*

*et al., 2021*). Our model predicts that children with immature TSPs should struggle with learning categories with more atypical exemplars or arbitrary features and should have poor memory for category exceptions. In line with these predictions, infants struggle to learn low coherence categories (*Gómez and Lakusta, 2004*; *Younger, 1990*; *Younger and Gotlieb, 1988*), and young children demonstrate poorer memory for exceptions than typical category members (*Savic and Sloutsky, 2019*). Our model may resolve the puzzle in the developmental literature about the discrepancy between infants' precocious performance on categorization tasks, on the one hand, and poor episodic memory abilities, on the other hand (*Keresztes et al., 2018*). To the extent that infants have access only to the MSP, our model predicts poor recognition performance (especially for atypical category instances) but intact categorization and even enhanced generalization. A fully operating basic hippocampal circuitry is eventually needed for learning low-coherence categories and for mature episodic memory functions which emerge later in development (*Gómez and Edgin, 2016*).

## Neuropsychological accounts of hippocampal contributions to category learning

Initial accounts of the role of the hippocampus in category learning came from studies of patients with MTL damage. Patients have been tested on a range of category learning tasks, including random dot patterns, probabilistic categories, faces, scenes, and painting categorization (*Kéri et al., 2001*; *Knowlton and Squire, 1993*; *Kolodny, 1994*; *Reber et al., 1996*; *Reed et al., 1999*; *Zaki et al., 2003*). *Knowlton and Squire, 1993* tested amnesics' ability to learn abstract novel categories of random dot patterns and observed similar categorization performance as in healthy controls, but impaired recognition, leading to the proposal that the MTL is not involved in category learning. However, amnesics do show impairment on a more difficult version of this task (learning categories A vs. B, as opposed to simply A vs. not-A; *Zaki et al., 2003*). Amnesics are also impaired on categorizing paintings by artist (*Kolodny, 1994*), and while they succeed in a categorization task with faces, they fail with scenes (*Graham et al., 2006*). Alzheimer's patients also show intact performance on the A/not-A task (*Kéri et al., 2001*; *Zaki et al., 2003*) but poor performance on the A/B task (*Zaki et al., 2003*), and categorization performance deteriorates as the disease progresses (*Kéri et al., 2001*). Rats with hippocampal damage also show impaired visual categorization (*Kim et al., 2018*). Overall, MTL damage leaves some ability to learn novel categories, indicating that the hippocampus is not the only region involved in category learning, but also creates clear deficits, especially when aggregating evidence across studies (*Zaki, 2004*), indicating that the hippocampus makes a causal contribution. Our proposal is that the MSP is responsible for that hippocampal contribution. This predicts that patients with specific TSP damage should be unimpaired in category learning, consistent with recent evidence of preserved statistical learning performance in a patient with specific DG damage (*Wang et al., 2023*).

## Neural evidence of hippocampal involvement in category learning

Neuroimaging studies provide strong additional evidence for hippocampal involvement in category learning and generalization (*Bowman and Zeithamova, 2018*; *Kumaran et al., 2009*; *Mack et al., 2016*; *Mack et al., 2018*; *Mack et al., 2020*; *Zeithamova et al., 2008*). There is also a neurophysiological literature on concept / category representation in the human hippocampus, with demonstrations of cells that respond similarly to distinct instantiations of a particular concept (e.g., Jennifer Aniston; *Quiroga et al., 2005*) and cells that respond invariantly across exemplars of higher level categories (e.g., faces; *Kreiman et al., 2000*). These findings are consistent with our proposal that the hippocampus contains representations that exhibit invariance across exemplars. Hippocampal subfields have not generally been investigated directly in neural studies of category learning, with one exception being our finding that CA1 but not CA3/DG represented category structure in the satellite stimuli (*Schapiro et al., 2018*). However, anterior hippocampus has a much larger proportion of the CA1 subfield than posterior hippocampus in humans (*Poppenk et al., 2013*; *Dalton et al., 2019*), so our account predicts that anterior hippocampus should preferentially reflect the learning of category structure. Indeed, several studies have found that activation in the anterior hippocampus is related to category learning (*Mack et al., 2016*; *Mack et al., 2018*; *Zeithamova et al., 2008*) and to prototype-style learning specifically (*Bowman and Zeithamova, 2018*). Further, activation in the hippocampal body and tail is associated with learning categories that involve exceptions (*Davis et al.,*

*2012*), which is also consistent with the finding that TSP white matter integrity predicts exception learning (*Schlichting et al., 2021*). There are many differences, however, between anterior and posterior hippocampus apart from subfield ratios (e.g., differential connectivity and tuning to spatial scale), so future work will be needed to more directly test these connections between subfield properties and the properties of anterior and posterior hippocampus.

### Recruitment of multiple neural systems during rapid category learning

As described above, we know that the hippocampus is not the only region contributing to category learning, with the basal ganglia and various regions of the neocortex known to be critically involved (*Ashby and Maddox, 2005*; *Seger and Miller, 2010*). While the hippocampus and basal ganglia seem especially important for rapid category learning, on the timescale of minutes to hours, most cortical regions likely support slower learning, across days, weeks, and months (*Seger and Miller, 2010*). An important exception is the prefrontal cortex, which is critical for certain kinds of rapid category learning, especially when working memory is required (*Ashby and O'Brien, 2005*). The involvement of the hippocampus relative to basal ganglia seems to hinge in part on the presence of feedback. The feedback version of the Weather Prediction Task preferentially engages the basal ganglia, whereas the observational version engages the MTL (*Poldrack et al., 2001*), and patients with Parkinson's disease (affecting basal ganglia function) show deficits in the feedback version but preserved performance in the observational version (*Shohamy et al., 2004*). Our view, based on these findings as well as literatures outside the domain of category learning, is that the hippocampus (and the MSP in particular) should be especially important in situations that involve more neutral, observational learning, with less motor response, feedback, or reward. Our current model has provided an account of the potential isolated contributions of the hippocampus to category learning, and future extensions should explore interactions with the basal ganglia, prefrontal cortex, and other cortical areas.

### Conclusions

We have put forward an account of the possible contributions of the hippocampus to rapid, novel category learning. We propose that the TSP, known for its rapid binding and pattern separation computations, contributes to remembering the arbitrary aspects of categories—the specifics or atypical aspects of individual exemplars or observations. The MSP, with a relatively slower learning rate and more distributed representations, contributes to learning the systematic aspects of categories—the structure shared across category exemplars. This proposal for two systems within the hippocampus with complementary expertise makes specific predictions. First, the neural responses to exemplars from the same category should tend to be more similar in CA1 than in CA3 and DG. This should be especially true immediately after presentation of a stimulus, as the circuit is recurrent, so information in CA1 can spread through EC back to DG and CA3 with more processing time (e.g., as in *Figure 3f*). Second, variation in the integrity of the two pathways—through individual differences, development, aging, psychiatric disorders, or neurological disease—should have differential behavioral consequences. Disruptions to TSP function should result in poor memory for specific, arbitrary features of exemplars but preserved memory for structure shared across exemplars, whereas disruptions to MSP function should result in poor memory for shared structure and relatively preserved memory for specifics. We predict behavioral consequences to be stronger in paradigms that involve more passive, observational learning, as the basal ganglia is more likely to be able to pick up the slack in tasks involving motor responses and feedback. There are several empirical datapoints that already fit these predictions, including category-related similarity structure in CA1 (*Schapiro et al., 2016*), TSP white matter integrity predicting exception learning (*Schlichting et al., 2021*), and behavioral exception learning that unfolds in accordance with MSP and TSP properties (*Heffernan et al., 2021*). But more work is needed to establish the extent to which this account correctly characterizes the contribution of the hippocampus to category learning. We hope this work inspires new theoretically diagnostic empirical studies as well as further model development.

### Acknowledgements

We are grateful to Siri Krishnamurthy, Brynn Sherman, and Dhairyya Singh for helpful discussions and assistance. This work was supported by a St Hugh's College Travel Grant, University of Oxford to JS, and Charles E Kaufman Foundation KA2020-114800 to ACS.

# Additional information

## Competing interests

Anna C Schapiro: Reviewing editor, *eLife*. The other author declares that no competing interests exist.

## Funding

| Funder | Grant reference number | Author |
|---|---|---|
| University of Oxford | St Hugh's College Travel Grant | Jelena Sučević |
| Charles E. Kaufman Foundation | KA2020-114800 | Anna C Schapiro |

The funders had no role in study design, data collection and interpretation, or the decision to submit the work for publication.

## Author contributions

Jelena Sučević, Conceptualization, Data curation, Formal analysis, Funding acquisition, Visualization, Methodology, Writing – original draft, Writing – review and editing; Anna C Schapiro, Conceptualization, Formal analysis, Supervision, Funding acquisition, Methodology, Writing – original draft, Writing – review and editing

## Author ORCIDs

Jelena Sučević ⓘ https://orcid.org/0000-0001-5091-5434
Anna C Schapiro ⓘ http://orcid.org/0000-0001-8086-0331

## Decision letter and Author response

Decision letter https://doi.org/10.7554/eLife.77185.sa1
Author response https://doi.org/10.7554/eLife.77185.sa2

---

# Additional files

## Supplementary files

• Supplementary file 1. Supplementary material includes input pattern and parameter details.

• Transparent reporting form

## Data availability

The current manuscript is a computational study, so no empirical data have been generated for this manuscript. The model is available on GitHub (copy archived at *Schapiro Lab, 2022*). Source data and source code for Figures 3, 4 and 5 are provided.

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
