## [Editor Report]

This article will be of interest to a broad audience of cognitive neuroscientists interested in learning and memory, especially those who study the computations of the hippocampus in human and animal models. This work offers compelling evidence in support of a role for the computations theorized to occur within the hippocampus in category learning more generally. The well-conducted and rigorous computational simulations support the key conclusions and offer a novel theoretical entry into characterizing human learning.

---

## [Decision Letter]

**Decision letter after peer review:**

Thank you for submitting your article "A neural network model of hippocampal contributions to category learning" for consideration by *eLife*. Your article has been reviewed by 3 peer reviewers, one of whom is a member of our Board of Reviewing Editors, and the evaluation has been overseen by Michael Frank as the Senior Editor. The reviewers have opted to remain anonymous.

Essential revisions:

1. Distinction from Schapiro et al., 2017: It is key to distinguish the current work from the simulations and findings in Schapiro et al. (2017). Although Reviewers were convinced that demonstrating that C-HORSE naturally accounts for category learning across a broad range of categorization tasks is novel and a worthy contribution, but how this is different from the senior author's prior work is not well argued in the current manuscript. In particular, the authors should address the conceptual differences between statistical/inferential learning (as is the focus in the 2017 paper) and category learning to highlight the novelty of the current work.

2. Apparent disconnect with established findings from unit recordings in CA1 and CA3: One concern, best described by Reviewer 2, is that in accounting for both statistical and category learning effects, C-HORSE may be unable to account for the more well-established body of empirical findings from unit recordings of hippocampal subfields. For example, it is not clear if the type of place and concept coding in hippocampal cells from rodents and humans are amenable to the predictions of C-HORSE. The Reviewers thought that this should be directly addressed by reviewing the literature which describes the response of single cells in CA1 and CA3 and considering how this corresponds to the predictions of the model, noting limitations where appropriate. Relatedly, Reviewer 3 noted that although the discussion of the CA1 vs. CA3 as it relates to functional differences in anterior vs. posterior hippocampus is an interesting point, the authors should soften their language here. Certainly, the C-HORSE findings coupled with anterior-posterior differences in subfields offers a compelling avenue for reconciling these viewpoints, but the matter is not as resolved as the discussion currently implies.

3. Situating C-HORSE in the literature: As a neurobiologically inspired model that provides insight into higher-level cognition, C-HORSE is broadly relevant to several research domains and existing theoretical frameworks (e.g., CLS, formal models of category learning, etc.). However, the Reviewers felt that it was not clear how to best place the proposed model in the literature. A formal comparison of C-HORSE to extant models seems beyond the scope of the current work. But, as a proof-of-concept alternative framework, the current work demonstrates how a single brain structure (i.e., hippocampus) can support both memory generalization and specificity. As such, the Reviewers suggest that making this proof-of-concept aspect explicit will help resolve confusion as to how C-HORSE in its current state should be considered alongside related theories/models.

4. Clarifying claims: In discussing the implications of their findings, the authors make several claims that over generalize their findings. For example, it is noted multiple times that MSP is "critical" and "responsible" for detecting regularities that support category generalization. It is true that MSP is clearly supporting this sort of generalization and more so than TSP, yet the simulation results also clearly show that the TSP-only model is still capable of above-chance categorization. The Reviewers suggest that the authors revise these statements to better align with the findings.

5. Directly characterizing the nature of representations in simulated tasks: The RSA approach is leveraged only in simulation 1, but would be helpful to consider for the other two simulations as well. In particular, many of the general versus specific claims made are based on indirect inferences from learning measures, when a direct characterization of the representations and how they change over learning could be made with RSA. The authors should consider adding these analyses for all simulations to better support their conclusions or provide a rationale for why they are not necessary.

6. Logic of initial vs. settled representations: In the RSA results of simulation 1, initial and settled representations are presented and compared, yet there is no logic provided as to why this is an important comparison to make (or even what initial vs. settled representations are, see point 7 below). The authors should provide a rationale for this analysis in terms of the learning mechanisms and information flow in the model.

7. Relationship to human learning findings: For each simulation, the qualitative fit between C-HORSE and end-of-learning behaviour from the prior work is mentioned in the main text to demonstrate a qualitatively "good fit" between model and human. Reviewer 1 suggested that these comparisons are expanded to (1) include behavioural measures across learning where appropriate and (2) include depictions of these behavioural effects in the figures. To be clear, quantitative fits of the model to empirical data are not expected, but that learning trajectories and end of learning performance in both model and human participants are more thoroughly considered in the text and figures.

8. Details of model and modelling approach: Although C-HORSE has been described in more detail in a prior paper (Schapiro et al., 2017), the Reviewers felt that more of these details should be included in the current work, especially since this will potentially reach an audience unfamiliar with the originating paper. In particular, important model mechanisms like learning rates, unit numbers across the layers, rationales for differences in TSP and MSP weights, cycles, and clamping should be further described and motivated.

9. More focused discussion: Although the discussion offers a comprehensive view on the role of hippocampus in category learning more broadly, it is not always connected back to the main conclusions of the paper. Reviewers suggested streamlining the discussion to include only those sections most relevant (see Reviewers 1 and 2 for specifics).

10. Clarity suggestions: The Reviewers also had several other suggestions that might increase the clarity of the methods, results, and discussion. The authors should please consider these suggestions and implement if they see fit.

*Reviewer #1 (Recommendations for the authors):*

I think this work is very exciting and compelling. However, I am certainly an insider in this field and am familiar with category learning research in general and relating hippocampal-based memory functions to learning behaviour. As such, it took a few reads to realize that the manuscript as is perhaps assumes to much knowledge of the reader. I think the contribution could be greatly strengthened if:

– More model details were provided. A citation is provided to the Schapiro et al., 2017 study, but important elements of the model that speak to key learning constructs are omitted (e.g., what is a cycle? what are initial and settled representations?)

– The authors should consider either directly evaluating the predictive power of C-HORSE relative to other models or recognizing the need for such an evaluation in future work as an important point for the discussion.

– Overall, the discussion could be refocused (and likely shortened) to put greater emphasis on the implications of the current findings to the broader themes in the literature.

– The RSA approach utilized in simulation 1 to characterize subfield representations should be used for all simulations potentially for both intact and lesion-variants of the model. Although the authors' main conclusions (MSP=detecting regularities, TSP=encoding items) can be inferred from the generalization and recognition performance of the lesion simulations, adding a more detailed and direct exploration of the model would strengthen the contribution significantly.

– The behaviour signatures of the original studies were described and depicted. Although there is some effort to describe how each of the three tasks capture distinct components of category learning, more description of these original studies in terms of their key behavioural patterns and what they reveal about category learning would be helpful. End of learning behavioural performance is sometimes provided in the main text, but it would help clarify the degree of fit from C-HORSE if these average accuracy measures from the prior work were plotted alongside the model results.

– In systematically varying the typicality of exemplars, the third simulation offers an interesting testbed for characterizing the contribution of MSP and TSP. And, in the analyses provided, there are hints at this. Recognition is better for TSP than MSP with increasing atypical exemplars. And, it is compelling that MSP matches TSP recognition with 1-2 atypical features. What I found most intriguing about this simulation is that the intact model offers the best performance. How is the information from both pathways combined in the model to drive good recognition? Characterizing this aspect of the model dynamics would potentially provide some insight into how the so-called complementary hippocampal functions are actually complementary.

– Relatedly, although the simulation results are interesting in this third study, I was left wondering how well they matched human performance. As suggested above, demonstrating the degree that C-HORSE actually matches human performance is key to understanding how well this new model truly accounts for human learning. As is, it is difficult to evaluate with this explicit comparison.

*Reviewer #2 (Recommendations for the authors):*

This model does come from a long 'lineage of models developed to account for episodic memory phenomena', and those should be more extensively cited (rather than only including papers authored or co-authored by Randy O'Reilly). I would suggest Marr (1971) Phil Trans B at the very least, and I think Gluck and Myers (1993) Hippocampus is also particularly relevant

The Discussion is far too long (thirteen and a half pages, about half of the total length of the manuscript). Please try and reduce the word count by sticking to the most pertinent issues

The authors state that the "monosynaptic pathway … was responsible for detecting the regularities that define category structure" and, later, that "the MSP was critical for learning the regularities underlying category structure and was responsible for generalization of knowledge to novel exemplars", but their results show that simulations with the TSP alone consistently perform better than chance on tests of either function (e.g. Figure 3A-C, 4B, 5B). As such, these statements appear to misrepresent the results. Please clarify

*Reviewer #3 (Recommendations for the authors):*

1. The one analysis that seems missing is analysis of generalization in Task 3 based on typicality. It would be informative, especially given that the training data showed interesting dissociations based on typicality.

2. I did not understand the relationship between time and performance during test in Task 1 and Task 3, where there are distinct training and test phases. I thought there are no labels and no weight adjustments at this stage. Why is the already trained network starting at chance at generalization test and then improve? How can we reconcile it with human performance that does not show such test accuracy pattern?

3. The clarity and flow of writing was exceptional, further enhancing interesting content, making this one of my favorite reviews this year. I did find a few challenging sentences, which perhaps could use rewording for clarity. I also found a couple of details that I would like clarified.

– Lines 66-69 could be split into two sentences, one defining complementary learning systems, another noting it may exist within hippocampus itself in distinct pathways. As written, it was confusing.

– Consider whether TSP and MSP abbreviations are necessary or if the words trisynaptic and monosynaptic could be spelt out each time instead (I am aware of the frequency, so this is just for consideration).

Methods details:

– Line 149 could add rationale for the distinct number of units in the different hidden layers

– Line 151 could mention the weight constraints

– Line 173 could explain "clamped" in non-technical terms

– Line 333/388 could explain why each category was represented by 2 units/5 units

– The additional task visualizations for task 2 and task 3 in Figure 2 were very helpful (the outcomes and %chance for cards combinations, the feature value visualization using the circles in task 3). Perhaps they could be explained more in the legend.

4. Connection to other work:

Line 53 puts up the idea that hippocampus may be well suited for category learning after all. It may be worth referencing a couple recent review papers that made the same point (e.g., Mack et al. 2018; Zeithamova and Bowman, 2020).

Line 293-296 Big Loop Recurrence could reference Koster et al., 2018.

[Editors' note: further revisions were suggested prior to acceptance, as described below.]

Thank you for resubmitting your work entitled "A neural network model of hippocampal contributions to category learning" for further consideration by *eLife*. Your revised article has been re-reviewed by one of the original reviewers and evaluated by Michael Frank (Senior Editor) and a Reviewing Editor.

We found that you were highly responsive to the previous comments and the manuscript has been significantly improved. There are only a couple of remaining issues that should be addressed, as outlined below:

1. The representation of exemplars and common features in your network could be made more intuitively understandable with the help of an illustration. For example, you could add another row to Figure 2A (or possibly 2B) that shows activity in the input layer to EC_in (which is clamped throughout learning), illustrating the activity pattern for each exemplar in one category. This is currently given in Supplementary Table 1A, but it would be useful to have it in the main text. Illustration of just one category would be sufficient to illustrate the pattern of activity across exemplars and perhaps also get a better idea of how the "classification" performance is evaluated in the network. This would complement the symbolic illustration of the exemplars, unique features and shared features from the three categories this is already a helpful part of Figure 2A. Alternatively, more information can be provided in the text.

2. The category structure effects are relatively subtle in Figure 4e and would benefit from a summary representation that more explicitly highlights their presence or absence in the different subfields. Adding a visual or numerical summary report of within-category vs. between-category similarity would be beneficial.

3. The Koster et al., 2018 citation was only added in the response letter but not the revised manuscript.

---

## [Author Response]

Essential revisions:1. Distinction from Schapiro et al., 2017: It is key to distinguish the current work from the simulations and findings in Schapiro et al. (2017). Although Reviewers were convinced that demonstrating that C-HORSE naturally accounts for category learning across a broad range of categorization tasks is novel and a worthy contribution, but how this is different from the senior author's prior work is not well argued in the current manuscript. In particular, the authors should address the conceptual differences between statistical/inferential learning (as is the focus in the 2017 paper) and category learning to highlight the novelty of the current work.

We agree that the novelty of the category learning simulations needed better articulation. We have added the following:

Introduction: “In the present work, we ask what computational properties of the hippocampus might allow it to contribute to category learning. Using a neural network model of the hippocampus named C-HORSE (Complementary Hippocampal Operations for Representing Statistics and Episodes), we previously demonstrated how the hippocampus might contribute to learning temporal regularities embedded in continuous sequences of stimuli (temporal statistical learning) and to inference over pairwise associations (Schapiro, Turk-Browne, et al., 2017; Zhou, Singh, Tandoc, and Schapiro, 2021). We showed that the heterogeneous properties of the two main pathways within the hippocampus may support complementary learning systems, with one pathway specializing in the rapid encoding of individual episodes and another in extracting statistics over time. This division of labor is analogous to the roles of the hippocampus and neocortex in the classic Complementary Learning Systems framework (McClelland, McNaughton, and O’Reilly, 1995), and our proposal was thus that a microcosm of this memory systems dynamic plays out within the hippocampus itself.

Category learning is related to temporal statistical learning in requiring information to be integrated across experiences, with the structure of a category discovered across exposure to individual exemplars. However, category learning is also different from temporal statistical learning in fundamental ways. Category learning involves tracking exemplars composed of separate features that can vary in different ways across exemplars. The regularities in these features often manifest in co-occurrence in space at one moment (e.g., different parts of an object), as opposed to co-occurrence nearby in time. There is also often demand in category learning tasks for more explicit grouping and labeling of exemplars. The present work evaluates to what extent the principles of structure learning that allow the hippocampus to support statistical learning may also apply to this different learning domain. If the principles generalize, it would suggest the possibility of broad, domain-general learning mechanisms at work in the hippocampus that allow integration of varied forms of information across experiences.”

While we see the demonstration of convergence across domains as novel, significant, and nontrivial, our expanded analyses and results in this version of the manuscript also unpack various findings from the model that are specific to category learning, which add important new insight into hippocampal contributions to category learning *in particular*.

2. Apparent disconnect with established findings from unit recordings in CA1 and CA3: One concern, best described by Reviewer 2, is that in accounting for both statistical and category learning effects, C-HORSE may be unable to account for the more well-established body of empirical findings from unit recordings of hippocampal subfields. For example, it is not clear if the type of place and concept coding in hippocampal cells from rodents and humans are amenable to the predictions of C-HORSE. The Reviewers thought that this should be directly addressed by reviewing the literature which describes the response of single cells in CA1 and CA3 and considering how this corresponds to the predictions of the model, noting limitations where appropriate.

We appreciate the encouragement to unpack the relationships between our model and the electrophysiology literature on properties of CA1 and CA3 cells. We view the model as strongly consistent with this literature, so it is useful for us to highlight these connections. First, in the place cell literature, CA1 spatial representations exhibit overlap across environments that reflects the similarity of those environments, whereas CA3 representations orthogonalize responses even for very similar environments, which is exactly the behavior of our model.

Introduction: “The projections within the TSP are sparse, enabling the formation of orthogonalized representations even with highly similar input patterns (i.e., pattern separation). This corresponds to the observed physiology; for example, distinct sets of place cells in rodents are responsive to particular locations in CA3 even in very similar enclosures (Leutgeb et al., 2004). The TSP is highly plastic in the brain and in the model, which supports rapid, even one-shot learning (Nakashiba et al., 2008). The MSP connects EC directly to CA1. These projections do not have the specialized sparsity of those in the TSP, allowing for more overlapping representations to emerge. Place cell responses in CA1 tend to overlap as a function of the similarity of the enclosure (Leutgeb et al., 2004). In addition, the MSP seems to learn more slowly (Lee, Rao, and Knierim, 2004; Nakashiba, et al., 2008).”

Our work can be viewed as studying how the hippocampus learns new concepts, which also connects it to the literature on concept cells in the human hippocampus. These cells respond to a particular concept, like Jennifer Aniston, and can be activated by her name or by different views of her face. Reviewer 2 points out that these neurons do not fire in response to other actors from Friends, which seems at odds with the units in our model that respond to all members of a category.

The understanding of concept cells is probably at too early a stage to say with confidence how they correspond to our proposal. One possibility is that concept cells actually reflect episodic memory. The experimenters in the concept cell studies conduct interviews with the patients prior to recording to determine their knowledge of celebrities, and the cells may thus correspond to memories of that prior conversation, which can be evoked by various different cues to the celebrity. This is related to an idea we put forward in a commentary in Trends in Cognitive Sciences (Solomon and Schapiro, 2022) that concept cells may reflect pattern completion of a concept, and would be consistent with the functions of our model’s TSP.

In the scenario where concept cells reflect semantic knowledge more than episodic memory, though, they still correspond to a kind of category. A cell could in theory be selective to a particular exemplar of Jennifer Aniston (e.g. her role in one movie) or it could correspond to *all* instances of Jennifer Aniston — defining a Jennifer Aniston *category*. Concept cells are thus categorical in this sense. There are also cells in the hippocampus that respond generally to faces and other higher-level categories (Kreiman et al., *Nature Neuroscience*, 2000), reflecting a superordinate category representation. Our MSP tends to cluster related experiences into overlapping neural ensembles, and would have similar representations in CA1 for different instances of Jennifer Aniston, as well as similar representations for different faces.

So while it is true that the studied Jennifer Aniston neurons do not fire in response to other actors from friends, given the presence of general face-selective cells, there are certainly *other* cells that would respond to multiple actors from friends. In sum, there are cells that reflect different levels of category structure in the hippocampus (reflecting our hierarchical understanding of categories).

Exploring these points seems out of the scope of the current paper, as there are multiple paths and they are all speculative at this point (and we need to cut down as opposed to expand our Discussion), but we do think it is important to acknowledge the potential connection to this literature. We have added the following to the Discussion:

“In addition to the neuroimaging literature on category learning, there is also neurophysiological literature on concept / category representation in the human hippocampus, with demonstrations of cells that respond similarly to distinct instantiations a particular concept (e.g., Jennifer Aniston; Quiroga et al., 2005) and cells that respond invariantly across exemplars of higher level categories (e.g. faces; Kreiman et al., 2000). These findings are consistent with our proposal that the hippocampus contains representations that exhibit invariance across exemplars.”

Relatedly, Reviewer 3 noted that although the discussion of the CA1 vs. CA3 as it relates to functional differences in anterior vs. posterior hippocampus is an interesting point, the authors should soften their language here. Certainly, the C-HORSE findings coupled with anterior-posterior differences in subfields offers a compelling avenue for reconciling these viewpoints, but the matter is not as resolved as the discussion currently implies.

We agree that the anterior vs. posterior hippocampus discussion needed qualification. We deleted a sentence in the Discussion that suggested that anterior/posterior differences provide an indirect test of the theory, and added the following to that paragraph:

“There are many differences, however, between anterior and posterior hippocampus apart from subfield ratios (e.g., differential connectivity and tuning to spatial scale), so future work will be needed to more directly test these connections between subfield properties and the properties of anterior and posterior hippocampus.”

3. Situating C-HORSE in the literature: As a neurobiologically inspired model that provides insight into higher-level cognition, C-HORSE is broadly relevant to several research domains and existing theoretical frameworks (e.g., CLS, formal models of category learning, etc.). However, the Reviewers felt that it was not clear how to best place the proposed model in the literature. A formal comparison of C-HORSE to extant models seems beyond the scope of the current work. But, as a proof-of-concept alternative framework, the current work demonstrates how a single brain structure (i.e., hippocampus) can support both memory generalization and specificity. As such, the Reviewers suggest that making this proof-of-concept aspect explicit will help resolve confusion as to how C-HORSE in its current state should be considered alongside related theories/models.

We agree that it is important for us to be clear that this model is a proof-of-concept for how the hippocampus may contribute to category learning and that formal comparison with other models is very valuable but out of scope of the present work. We have added the following sections to the Discussion:

“Our goal was to take a model with an architecture inspired by the anatomy and properties of the hippocampus and explore how it might accomplish category learning. While we did not endeavor to build in any particular strategies, the emergent behaviors of the model bear resemblance to existing psychological models of categorization. The model thus provides a bridge across levels of analysis, showing how neurobiological mechanisms may give rise to some of the more abstract operations of existing models.”

“There are many models of category learning in the literature, including neural network models that would likely exhibit behavior closely analogous to our model’s MSP. Our goal here is not to claim that our model better fits empirical data than existing models, but rather to provide a proof-of-concept demonstration of how the computations of hippocampal subregions may give rise to different components of category learning. Detailed comparison of the model’s behavior to other models in the literature will be valuable in evaluating and refining the model, however, and will be an important goal for future work.”

4. Clarifying claims: In discussing the implications of their findings, the authors make several claims that over generalize their findings. For example, it is noted multiple times that MSP is "critical" and "responsible" for detecting regularities that support category generalization. It is true that MSP is clearly supporting this sort of generalization and more so than TSP, yet the simulation results also clearly show that the TSP-only model is still capable of above-chance categorization. The Reviewers suggest that the authors revise these statements to better align with the findings.

This is an excellent point, and we have revised our language to be more accurate about the differences in expertise across the pathways without making strong claims about necessity:

Abstract: “The monosynaptic pathway from entorhinal cortex to CA1, in contrast, specialized in detecting the regularities that define category structure”

Introduction: “The MSP played a central role in learning the regularities underlying category structure and excelled in generalizing knowledge to novel exemplars. The TSP also contributed to behavior across the tasks but with the opposite expertise, specializing in memory for the unique properties of exemplars.”

Results: “the MSP contributed relatively more to this categorization ability, whereas the TSP was better able to process individual combinations of cards”

Discussion: “Across paradigms, the MSP specialized in detecting the regularities that define category structure.”

5. Directly characterizing the nature of representations in simulated tasks: The RSA approach is leveraged only in simulation 1, but would be helpful to consider for the other two simulations as well. In particular, many of the general versus specific claims made are based on indirect inferences from learning measures, when a direct characterization of the representations and how they change over learning could be made with RSA. The authors should consider adding these analyses for all simulations to better support their conclusions or provide a rationale for why they are not necessary.

We completely agree that RSA visualizations would be useful for all simulations. We have run these analyses for Simulations 2 and 3, and Figures 4 and 5 have been updated to include these. The new RSAs are convergent with those for Simulation 1, showing more similarity for items in the same category in CA1 relative to DG and CA3. The RSA for Simulation 3 provides some new insight into the model’s behavior, showing the limits of pattern separation in the TSP, as described below.

6. Logic of initial vs. settled representations: In the RSA results of simulation 1, initial and settled representations are presented and compared, yet there is no logic provided as to why this is an important comparison to make (or even what initial vs. settled representations are, see point 7 below). The authors should provide a rationale for this analysis in terms of the learning mechanisms and information flow in the model.

We agree that this needed to be much clearer. We have added the following section to the Methods:

“Representational similarity analyses. To assess the nature of learned representations in the networks, we performed representational similarity analyses for each of the simulations during a test phase at the end of training. We used Pearson correlation to relate the patterns of activity evoked by presentation of different items. We analyzed representations separately in the ‘initial’ and ‘settled’ response in the intact network. The initial response captures the activation pattern once activity has spread throughout the network but before output activity in EC_out_ recirculates back to the input in EC_in_ — before there is an impact of “big-loop” recurrence (Kumaran and McClelland, 2012; Schapiro, Turk-Browne, et al., 2017). The settled response captures the fully settled pattern of activity including the influence of big-loop recurrence. Big-loop recurrence permits the representations in CA1 to influence those in DG and CA3, so separate analysis of the initial response allows cleaner assessment of the unique representational contributions of the different subfields.”

We also reiterate these points now in the first presentation of representational similarity results:

“To assess network representations, we performed representational similarity analysis for each hidden layer of the intact network at the end of training (140 trials). We captured the patterns of unit activities evoked by presentation of each satellite’s unique feature (for the 12 satellites with unique features). There was no structure in the representations prior to training (not depicted), and the representations that emerged with training revealed sensitivity to the category structure. This was particularly evident in CA1 (Error! Reference source not found.f), with items from the same category represented much more similarly than items from different categories. We separately analyzed the initial pattern of activity evoked by each feature (before there was time for activity to spread from EC_out_ to EC_in_), and the fully settled response. The initial response allows us to understand the separate representational contributions of the subfields, before CA1 activity has the potential to influence DG and CA3. In the initial response, there was no sensitivity at all to category structure in DG and CA3 — items were represented with distinct sets of units. This is a demonstration of the classic pattern separation function of the TSP, applied to this domain of category learning, where it is able to take overlapping inputs and project them to separate populations of units in DG and CA3. CA1 representations, on the other hand, mirrored the category structure, with overlapping sets of units evoked by items in the same category. This result is consistent with our neuroimaging findings using this paradigm, where CA1 was the only subfield of the hippocampus to show significant within versus between category multivoxel pattern similarity (Schapiro et al., 2018). The settled response revealed sensitivity to category structure in all three hidden layers, reflecting the influence of CA1 on the rest of the network after “big-loop” recurrence (Koster et al., 2018; Kumaran and McClelland, 2012; Schapiro, Turk-Browne, et al., 2017), though CA1 still showed the strongest response. All sensitivity to the category structure in this network was thus driven by the learned representations in CA1.”

7. Relationship to human learning findings: For each simulation, the qualitative fit between C-HORSE and end-of-learning behaviour from the prior work is mentioned in the main text to demonstrate a qualitatively "good fit" between model and human. Reviewer 1 suggested that these comparisons are expanded to (1) include behavioural measures across learning where appropriate and (2) include depictions of these behavioural effects in the figures. To be clear, quantitative fits of the model to empirical data are not expected, but that learning trajectories and end of learning performance in both model and human participants are more thoroughly considered in the text and figures.

We very much appreciate this suggestion, as we agree that including comparisons to behavior would significantly strengthen the paper. For Simulation 1, we re-analyzed our own prior data in order to generate timecourses showing performance for unique and shared features over the course of learning (we did not have intermittent tests of generalization). Figure 3a and b show the human behavior with learning curve fits, which are a nice qualitative match to the intact model performance (Figure 3c and 3d).

For Figure 4, we now include a timecourse of categorization performance for amnesics and controls for the weather prediction task (Figure 4a). We also added a plot in the same format showing intact performance in our model alongside an average of the MSP/TSP lesion, as a rough analogy to the amnesic condition. The results are again highly consistent.

For Figure 5, we have included two panels of behavioral results from Bowman, Iwashita, and Zeithamova (2020) showing typicality effects on behavior, across learning, and at the final test (Figure 5a and 5b). We have added our model results in a comparable format in Figure 5c and 5d, again showing a nice correspondence.

8. Details of model and modelling approach: Although C-HORSE has been described in more detail in a prior paper (Schapiro et al., 2017), the Reviewers felt that more of these details should be included in the current work, especially since this will potentially reach an audience unfamiliar with the originating paper. In particular, important model mechanisms like learning rates, unit numbers across the layers, rationales for differences in TSP and MSP weights, cycles, and clamping should be further described and motivated.

We agree that it is useful to have more motivation and details in this paper. We have added more details and explanations throughout the Methods section.

9. More focused discussion: Although the discussion offers a comprehensive view on the role of hippocampus in category learning more broadly, it is not always connected back to the main conclusions of the paper. Reviewers suggested streamlining the discussion to include only those sections most relevant (see Reviewers 1 and 2 for specifics).

We have very substantially revised the Discussion to streamline and connect the sections better to the main points of the paper. We provide more information on the changes in response to Reviewers 1 and 2 below.

10. Clarity suggestions: The Reviewers also had several other suggestions that might increase the clarity of the methods, results, and discussion. The authors should please consider these suggestions and implement if they see fit.

We appreciate all the suggestions and have implemented almost all of them, as detailed below.

Reviewer #1 (Recommendations for the authors):I think this work is very exciting and compelling. However, I am certainly an insider in this field and am familiar with category learning research in general and relating hippocampal-based memory functions to learning behaviour. As such, it took a few reads to realize that the manuscript as is perhaps assumes to much knowledge of the reader. I think the contribution could be greatly strengthened if:– More model details were provided. A citation is provided to the Schapiro et al., 2017 study, but important elements of the model that speak to key learning constructs are omitted (e.g., what is a cycle? what are initial and settled representations?)

We agree. See response to Editor points #6 and #8 above.

– The authors should consider either directly evaluating the predictive power of C-HORSE relative to other models or recognizing the need for such an evaluation in future work as an important point for the discussion.

We agree that this is an important point to make, and have added the following to the Discussion:

“There are many models of category learning in the literature, including neural network models that would likely exhibit behavior closely analogous to our model’s MSP. Our goal here is not to claim that our model better fits empirical data than existing models, but rather to provide a proof-of-concept demonstration of how the computations of hippocampal subregions may give rise to different components of category learning. Detailed comparison of the model’s behavior to other models in the literature will be valuable in evaluating and refining the model, however, and will be an important goal for future work.”

– Overall, the discussion could be refocused (and likely shortened) to put greater emphasis on the implications of the current findings to the broader themes in the literature.

We agree and have implemented significant refocusing and shortening of the Discussion.

– The RSA approach utilized in simulation 1 to characterize subfield representations should be used for all simulations potentially for both intact and lesion-variants of the model. Although the authors' main conclusions (MSP=detecting regularities, TSP=encoding items) can be inferred from the generalization and recognition performance of the lesion simulations, adding a more detailed and direct exploration of the model would strengthen the contribution significantly.

We agree and have added representational similarity analyses for all simulations in this version of the paper.

– The behaviour signatures of the original studies were described and depicted. Although there is some effort to describe how each of the three tasks capture distinct components of category learning, more description of these original studies in terms of their key behavioural patterns and what they reveal about category learning would be helpful. End of learning behavioural performance is sometimes provided in the main text, but it would help clarify the degree of fit from C-HORSE if these average accuracy measures from the prior work were plotted alongside the model results.

We really appreciate this suggestion and have added empirical behavioral trajectories as comparisons for all of our simulations. (See Editor point #7 above.)

– In systematically varying the typicality of exemplars, the third simulation offers an interesting testbed for characterizing the contribution of MSP and TSP. And, in the analyses provided, there are hints at this. Recognition is better for TSP than MSP with increasing atypical exemplars. And, it is compelling that MSP matches TSP recognition with 1-2 atypical features. What I found most intriguing about this simulation is that the intact model offers the best performance. How is the information from both pathways combined in the model to drive good recognition? Characterizing this aspect of the model dynamics would potentially provide some insight into how the so-called complementary hippocampal functions are actually complementary.

This is a very interesting point. This is indeed the only case where we found that the intact model performed better than an individual pathway on a particular task. We think that this suggests that the MSP is playing a “supportive” role for the episodic machinery of the TSP, a role that has been characterized in the prior work with this lineage of models. It is not clear to us, though, why this role is important in this simulation and not in others. While exploring this issue is very important, we think it will be a significant undertaking and out of the scope of the current work. We have added some additional comment on this issue when discussing this result:

“Unlike prior simulations, in this case the intact network exhibited better performance than the lesioned networks, suggesting that this task benefits from having both pathways intact. The TSP-only network performed better than the MSP only network, which was virtually unable to recognize atypical category members (3 or 4 atypical features), but showed somewhat better performance on items more similar to the prototype (1 or 2 atypical features). The MSP can thus contribute to atypical feature memory to some extent, when the item is overall very similar to the prototype. The more arbitrary the item, the more the TSP is needed. Even for arbitrary items, though, the TSP benefitted from the presence of the MSP (as indicated by higher performance for the intact network), suggesting that this may be a situation where the MSP plays an important supportive function.”

– Relatedly, although the simulation results are interesting in this third study, I was left wondering how well they matched human performance. As suggested above, demonstrating the degree that C-HORSE actually matches human performance is key to understanding how well this new model truly accounts for human learning. As is, it is difficult to evaluate with this explicit comparison.

We agree. We have now included figures depicting human performance in a study by Bowman, Iwastiha and Zeithamova, (2020) that used this category learning task and presented these results alongside model performance.

Reviewer #2 (Recommendations for the authors):This model does come from a long 'lineage of models developed to account for episodic memory phenomena', and those should be more extensively cited (rather than only including papers authored or co-authored by Randy O'Reilly). I would suggest Marr (1971) Phil Trans B at the very least, and I think Gluck and Myers (1993) Hippocampus is also particularly relevant

We had a more specific intention here to describe how we are building on a particular neural network architecture implementing properties of the subfields and pathways of the hippocampus. The Marr and Gluck and Myers papers are of course central to the history of hippocampal modeling of episodic memory, but we want to make a narrower point about the architecture used here. We have updated to these narrower statements:

Methods: “We adopted a neural network model of the hippocampus developed after a lineage of models used to explain how the DG, CA3, and CA1 subfields of the hippocampus contribute to episodic memory (Ketz et al., 2013; Norman and O’Reilly, 2003; O’Reilly and Rudy, 2001).”

Introduction: “C-HORSE comes from a lineage of models developed to explain how the subfields of the hippocampus support episodic memory (Ketz, Morkonda, and O’Reilly, 2013; Norman and O’Reilly, 2003; O’Reilly and Rudy, 2001).”

The Discussion is far too long (thirteen and a half pages, about half of the total length of the manuscript). Please try and reduce the word count by sticking to the most pertinent issues

We agree, and have now made significant cuts to the Discussion to streamline this section.

The authors state that the "monosynaptic pathway … was responsible for detecting the regularities that define category structure" and, later, that "the MSP was critical for learning the regularities underlying category structure and was responsible for generalization of knowledge to novel exemplars", but their results show that simulations with the TSP alone consistently perform better than chance on tests of either function (e.g. Figure 3A-C, 4B, 5B). As such, these statements appear to misrepresent the results. Please clarify

We completely agree that the language we had was too binary — both pathways show evidence of supporting both kinds of functions to some extent, but with different relative specializations. See response to Editor point #4 above.

Reviewer #3 (Recommendations for the authors):1. The one analysis that seems missing is analysis of generalization in Task 3 based on typicality. It would be informative, especially given that the training data showed interesting dissociations based on typicality.

This is a great suggestion. We now include results in Figure 5 that show generalization as a function of typicality. Generalization gets easier with increasing typically, for both the MSP and TSP. This relates to our new RSA results now also shown in Figure 5, where DG/CA3 does show some category structure for highly typical exemplars.

New text describing these results:

“Like human participants, the intact network improved in its categorization of novel items across training, with better performance for more prototypical items (Figure 5c). Figure 5c plots the initial training period, with 10 trials prior to each interim test, Figure 5d shows the typicality gradient at the end of this training, and Figure 5e shows generalization behavior over a longer training period, broken down by pathway. As in the prior simulations, the MSP-only network excelled in generalization, outperforming the intact network by the end of training across all typicality levels. The TSP-only network performed much worse, but still above chance, and was able to generalize quite well for highly prototypical items.”

“Visualization of the internal representations of the model after training provides insight into the behaviors of the two pathways. Figure 5g shows the similarity of the evoked activity of the items in this domain, with items arranged from most prototypical members of one category to most prototypical members of the other. As in the prior simulations, DG and CA3 represented the items more distinctly than CA1, and settling activity after big-loop recurrence increased similarity, especially in CA1. This simulation was unique, however, in that DG and CA3 showed clear similarity structure for the prototype and highly prototypical items. There is a limit to the pattern separation abilities of the TSP, and these highly similar items exceed that limit. This explains why, at high typicality levels, the TSP could be quite successful on its own in generalization (Figure 5e), and why it struggled with atypical feature recognition for these items (Figure 5f).”

2. I did not understand the relationship between time and performance during test in Task 1 and Task 3, where there are distinct training and test phases. I thought there are no labels and no weight adjustments at this stage. Why is the already trained network starting at chance at generalization test and then improve? How can we reconcile it with human performance that does not show such test accuracy pattern?

The “Trials” shown in these plots refer to trials of training, not trials of test. We stop the network after a certain number of training trials and run a test with no learning. So Trial 0 corresponds to no training, and thus chance performance. We have added clarification that trials correspond to training to all of the relevant figure legends.

3. The clarity and flow of writing was exceptional, further enhancing interesting content, making this one of my favorite reviews this year. I did find a few challenging sentences, which perhaps could use rewording for clarity. I also found a couple of details that I would like clarified.

We truly appreciate the Reviewer’s positive view overall on the writing.

– Lines 66-69 could be split into two sentences, one defining complementary learning systems, another noting it may exist within hippocampus itself in distinct pathways. As written, it was confusing.

We have split this up into two sentences:

“We showed that the heterogeneous properties of the two main pathways within the hippocampus may support complementary learning systems, with one pathway specializing in the rapid encoding of individual episodes and another in extracting statistics over time. This division of labor is analogous to the roles of the hippocampus and neocortex in the classic Complementary Learning Systems framework (McClelland, McNaughton, and O’Reilly, 1995), and our proposal was thus that a microcosm of this memory systems dynamic plays out within the hippocampus itself.”

– Consider whether TSP and MSP abbreviations are necessary or if the words trisynaptic and monosynaptic could be spelt out each time instead (I am aware of the frequency, so this is just for consideration).

We appreciate the suggested but have decided to retain the abbreviations given the frequency of their use.

Methods details:– Line 149 could add rationale for the distinct number of units in the different hidden layers

We have added the following:

“There are 400 units in DG, 80 units in CA3, and 100 units in CA1, while input and output layer size (Input, EC_in_ and EC_out_) varied as a function of the task. The hidden layer size ratios reflect the approximate ratios in the human hippocampus (Ketz et al., 2013).”

– Line 151 could mention the weight constraints

We are unsure exactly what the Reviewer has in mind but have clarified the learning schemes controlling the weights.

– Line 173 could explain "clamped" in non-technical terms

We have updated this paragraph to read:

“There is also a separate Input layer (not shown in Figure 1) with the same dimensionality as EC_in_, where external input was clamped (i.e., forced to take on particular values), allowing activity in EC_in_ to vary as a function of external input as well as EC_out_ activity. There are one-to-one non-learning connections between Input and EC_in_ and between EC_in_ and EC_out_.”

– Line 333/388 could explain why each category was represented by 2 units/5 units

Here are the edited versions:

“Each card was represented by one unit in the input and output, and each weather outcome (category) was represented by two units (increasing the relative salience of category information).”

“Each feature was represented by 2 units (one unit for each of the two possible feature values), and each category label was represented by 5 units (increasing the salience of category information relative to the many creature features).”

– The additional task visualizations for task 2 and task 3 in Figure 2 were very helpful (the outcomes and %chance for cards combinations, the feature value visualization using the circles in task 3). Perhaps they could be explained more in the legend.

The legend now reads:

“Figure 1. Overview of simulated category learning paradigms. (a) Satellite categories: Distinct categories of novel “satellites” consisting of unique and shared features (Schapiro, McDevitt, et al., 2017). (b) Weather Prediction Task: each abstract card is probabilistically related to a category (sun or rain), and on a given trial, category must be guessed from a simultaneously-presented set of one to three cards (Knowlton et al., 1994). The illustration shows the first two cards related to the “sun” category on 90% of the trials (and to “rain” on 10% of the trials), while a combination of the first three cards related to sun 79% of the time, and a fourth card viewed by itself associated with sun 15% of the time. (c) Intermixed categories with varying typicality: Categories where each item consists of 10 binary features. The two prototypes on opposite sides of the feature space have no features in common (depicted by all green versus all yellow features in the piecharts), and the rest of the exemplars have a varying number of features in common with the prototypes (Zeithamova et al., 2008, material adapted from Figure 1, Figure Copyright [2008] Society for Neuroscience).”

4. Connection to other work:Line 53 puts up the idea that hippocampus may be well suited for category learning after all. It may be worth referencing a couple recent review papers that made the same point (e.g., Mack et al. 2018; Zeithamova and Bowman, 2020).

Agreed, and added.

Line 293-296 Big Loop Recurrence could reference Koster et al., 2018.

We have added this indeed very relevant reference.

[Editors’ note: what follows is the authors’ response to the second round of review.]

We found that you were highly responsive to the previous comments and the manuscript has been significantly improved. There are only a couple of remaining issues that should be addressed, as outlined below:1. The representation of exemplars and common features in your network could be made more intuitively understandable with the help of an illustration. For example, you could add another row to Figure 2A (or possibly 2B) that shows activity in the input layer to EC_in (which is clamped throughout learning), illustrating the activity pattern for each exemplar in one category. This is currently given in Supplementary Table 1A, but it would be useful to have it in the main text. Illustration of just one category would be sufficient to illustrate the pattern of activity across exemplars and perhaps also get a better idea of how the "classification" performance is evaluated in the network. This would complement the symbolic illustration of the exemplars, unique features and shared features from the three categories this is already a helpful part of Figure 2A. Alternatively, more information can be provided in the text.

These are very helpful suggestions for clarifying the format of the model input. Figure 2A now has grids that illustrate the input for one of the satellite categories:

2. The category structure effects are relatively subtle in Figure 4e and would benefit from a summary representation that more explicitly highlights their presence or absence in the different subfields. Adding a visual or numerical summary report of within-category vs. between-category similarity would be beneficial.

While it is true that the Weather Prediction Task resulted in much noisier category representations than the other simulations (because of the probabilistic nature of the task), we would argue that the category effects are still readily visible by eye. In the CA1 initial response — the key representational similarity analysis for this simulation — it is clear that there are more dark blue squares outside than inside the black boxes. While we are wary of crowding the Results with too many new stats, we completely agree that it is important to reassure readers about the reliably of the category effects given the noisiness of the CA1 category structure relative to the other simulations. We have added the following means / stats:

“As in Simulation 1, DG and CA3 represented the card combinations relatively distinctly, whereas the patterns of activity in CA1 reflected the category structure (mean similarity within category: 0.38, across: 0.25; *p* <.001).”

3. The Koster et al., 2018 citation was only added in the response letter but not the revised manuscript.

Thank you so much for catching this error. The Koster citation is now in the text.